# Signatures of human intervention – or not? Downstream intensification of hydrological drought along a large Central Asian River: the individual roles of climate variability and land use change

Artemis Roodari[1,2], Markus Hrachowitz[1], Farzad Hassanpour[2] and Mostafa Yaghoobzadeh[3]

[1]Department of Water Management, Faculty of Civil Engineering and Geoscience, Delft University of Technology, Stevinweg 1, 2628CN Delft, Netherlands
[2]Department of Water Engineering, Faculty of Soil and Water Sciences, University of Zabol, Zabol, 98615-538, Iran
[3]Department of Water Engineering, University of Birjand, Birjand, 97175-615, Iran

*Correspondence to*: Artemis Roodari (artemis_roodari@yahoo.com)

**Abstract.** The transboundary Helmand River basin is the main drainage system for large parts of Afghanistan and the Sistan region of Iran. Due to the reliance of this arid region on water from the Helmand River, a better understanding of hydrological drought pattern and the underlying drivers in the region are critically required for effective management of the available water. The objective of this paper is therefore to analyse and quantify spatio-temporal pattern of drought and the underlying processes in the study region. More specifically we test for the Helmand River Basin the following hypotheses

for the 1970-2006 period: (1) drought characteristics, including frequency and severity systematically changed over the study period, (2) the spatial pattern and processes of drought propagation through the Helmand River Basin also changed and (3) the relative roles of climate variability and human influence on changes in hydrological droughts can be quantified.

It was found that drought characteristics varied throughout the study period, but did largely show no systematic trends. The same was observed for the time series of drought indices SPI and SPEI, which exhibited considerable spatial coherence and

synchronicity throughout the basin indicating that, overall, droughts similarly affect the entire HRB with little regional or local differences. In contrast, analysis of SDI exhibited significant negative trends in the lower parts of the basin, indicating an intensification of hydrological droughts. It could be shown that with a mean annual precipitation of ~250 mm y$^{-1}$, streamflow deficits and thus hydrological drought throughout the HRB are largely controlled by precipitation deficits, whose annual anomalies on average account for ±50 mm y$^{-1}$ or ~20% of the water balance of the HRB, while anomalies of total

evaporative fluxes on average only account for ±20 mm y$^{-1}$. Assuming no changes in the reservoir management practices over the study period, the results suggest that the two reservoirs in the HRB only played a minor role for the downstream propagation of streamflow deficits, as indicated by the mean difference between inflow and outflow during drought periods which did not exceed ~0.5% of the water balance of the HRB. Irrigation water abstraction had a similarly limited effect on the magnitude of streamflow deficits, accounting for ~10% of the water balance of the HRB. However, the downstream parts

of the HRB *moderated* the further propagation of streamflow deficits and associated droughts because of the minor effects of reservoir operation and very limited agricultural water in the early decades of the study period. This drought moderation function of the lower basin was gradually and systematically inverted by the end of the study period, when the lower basin

eventually *amplified* the downstream propagation of flow deficits and droughts. Our results provide plausible evidence that this shift from drought moderation to drought amplification in the lower basin is likely a consequence of increased

agricultural activity and the associated increases in irrigation water demand from ~13 mm $y^{-1}$ at the beginning of the study period to ~23 mm $y^{-1}$ at the end and thus in spite of being only a minor fraction of the water balance. Overall the results of this study illustrate that flow deficits and the associated droughts in the HRB clearly reflect the dynamic interplay between temporally varying regional differences in hydro-meteorological variables together with subtle and temporally varying effects linked to direct human intervention.

## 40    1 Introduction

There is evidence that droughts have the potential to increasingly affect human societies as well as ecosystem functioning. In a world under change, decision-makers therefore need reliable quantitative information about drought characteristics to ensure the development and implementation of effective and sustainable water management procedures. To be reliable this information needs to be based on a solid understanding of how different types of droughts propagate through different

hydrological systems. While meteorological droughts are controlled by precipitation deficits only, agricultural and hydrological droughts are caused by soil moisture and runoff deficits, respectively. As pointed out, amongst others, by Mishra and Singh (2010) the processes underlying droughts are complex because they are dependent on many interacting processes in terrestrial hydrological systems, such as the interaction between the atmosphere and the hydrological processes which feed moisture to the atmosphere. Therefore, monitoring and analysis of hydrological droughts have received increased

attention in recent decades (van Huijgevoort et al., 2014; Pathak and Dodamani, 2016; Weng et al., 2015; Vicente-Serrano et al., 2012; Kubiak-Wójcicka and Bak, 2018; Trambauer et al., 2014; Ahmadalipour et al., 2017; Jiao and Yuan, 2019; Moravec et al., 2019). In general, it is well-understood that both, agricultural and hydrological droughts are modulated by the interactions of climate, river basin characteristics, such as geology, as well as a human influence or any combination thereof (e.g. Van Lanen et al., 2013; Huang et al., 2016; Liu, et al., 2016; Van Loon, et al., 2019). For example, data show that

reservoir operations can have both, considerable positive or negative effects on downstream hydrological drought pattern (e.g. Zhang et al., 2013; Pingue et al., 2016; Wu et al., 2017), which may politically be particularly sensitive for transboundary rivers in arid envrionments (Al-Faraj and Scholz, 2015, Wan et al., 2018).

The transboundary Helmand River system between Afghanistan and Iran is the primary contributor of water to the Hamun lake- and wetland-system in the Sistan Plain, which is the terminus of one of the largest endorheic basins in Central Asia. In

this region, which is described as one of the driest, most remote deserts on Earth (Whitney, 2006), water from the Helmand River system plays a critical role not only to sustain agricultural production, hydropower generation and ecosystem stability but also for drinking water supply for some one million people living in the region, including the cities of Kandahar in Afghanistan and Zabol in Iran.

The area has recently experienced a severe, multi-year drought (1998–2004). Reduction of flow and episodic no-flow conditions in the Helmand River during this period have caused significant disruption of water supply. As a consequence, agricultural production dropped by almost 90% as compared to average no-drought conditions, further resulting in food shortage and considerable economic damage (Ebrahimzadeh and Esmaelnejad, 2013). Given the region's extreme dependence on water from the Helmand River system and the associated vulnerability to hydrological droughts, a few recent studies started to analyse droughts in Afghanistan and the Helmand River Basin (e.g. Ahmad and Wasiq, 2004; Miyan, 2015). For example, Alami et al. (2018) analyzed meteorological droughts in the Helmand River Basin using different methods and quantitatively documented the extreme drought in 2001. However, most of the research in this region focused on the application of hydrological models for the simulation of runoff to provide decision bases for integrated water management issues in the region. These studies include Hajihosseini et al. (2016), who assessed the Afghan-Iranian Helmand River Treaty (The Iranian-Afghan Helmand (Hirmand) River Water Treaty, 1973) using the SWAT model (Arnold et al., 1998) and data from the Climatic Research Unit (CRU; Harris et al., 2014). A study by Wardlaw et al. (2013) formulated a model for the development of water resources systems in the Helmand River Basin using the Water Evaluation and Planning (WEAP) model and established a list of scenarios for the future.

Similarly, Vining and Vecchia (2007) estimated future runoff conditions of the river to evaluate the effects of different reservoir operation strategies under different climate change scenarios on downstream water supply. Van Beek et al. (2008) developed methods and tools to build the capacity to sustain agriculture and ecosystems in the downstream Sistan Plain. In spite of this growing body of literature for the region, the scarcity of reliable meteorological and hydrological data so far limited systematic, quantitative analysis of spatio-temporal pattern of hydrological droughts and the underlying drivers and processes in the Helmand River Basin.

Due to the reliance of the region on water from the Helmand River, a better understanding of hydrological drought pattern and the underlying processes in the region are critically required for effective management of the available water. Most studies in the Helmand River Basin have so far remained limited to mere documentation and/or general assessments of mostly meteorological drought characteristics. We here will extend this scope also to hydrological drought and evaluate the meteorological drought under the additional role of atmospheric water demand. The overall objective of this paper is therefore to analyse and quantify changes in spatio-temporal pattern of drought charaterisitics and the underlying processes in the study region in an attempt to quantitatively attribute these changes to climate and human interventions, respectively. More specifically we will test for the Helmand River Basin the following hypotheses for the 1970-2006 period: (1) drought characteristics, including frequency and severity systematically changed over the study period, (2) the spatial pattern and processes of drought propagation through the Helmand River Basin also changed and (3) the relative roles of climate variability and human influence on changes in hydrological droughts can be quantified.

## 2 Study area

The endorheic Helmand River Basin (HRB; Figure 1) covers approximately 105,000 km$^2$ or 15% of Afghanistan. From its source area, in the Koh-i-Baba mountains, an extension of the Hindu Kush west of Kabul, with elevations to over 4600 masl, the Helmand River system drains into the Hamun lake and wetland system in the Sistan plain of Eastern Iran, a closed inland delta with a minimum elevation of 440 masl in the south-west of the HRB, which covers 5 % of the total HRB area (Goes et al., 2016). Both, long-term mean annual precipitation ($\bar{P}$=90-480 mm yr$^{-1}$; Figure 1d) and potential evaporation ($\overline{E_P}$ =700-1800 mm yr$^1$; Figure 1e) exhibit considerable spatial variability throughout the HRB. This results in a pronounced gradient of aridity from sub-arid in the North-East to hyper-arid conditions in the South-West as expressed by the aridity index I$_A$ ($I_A = \frac{\bar{P}}{\overline{E_P}}$ [-]; Figure 1f). Precipitation falls mostly in the winter months and in the upper basin almost always occurs as snow. In general, snowmelt generates the annual runoff peaks in early spring and sustains flow in the HRB throughout the dry summers. For the following analysis, the HRB is divided into six sub-basins (Figure 1c; Table 1): the Upper Helmand River Basin (UHRB) with the main stem of the Helmand River, the Central Helmand River Basin (CHRB) and the Upper Arghandab River Basin (UARB) as well as the Lower Arghandab River Basin (LARB) are nested in and drain into the Lower Helmand River basin (LHRB) and subsequently into the Sistan plain (SISP). The UHRB accounts for 80% of the combined inflow into the LHRB. Flow in the LHRB is influenced by the operation of two upstream reservoirs (Figure 1b; Table 1). While the reservoir at Kajakai Dam with a storage capacity of 1800 mio. m$^3$, located at the outflow of the UHRB, is a multi-purpose structure for electricity production, flood control and irrigation water supply, the smaller Dahla Dam, located at the outlet of the UARB into the LARB about 180 km upstream of the confluence with the LHRB, has a storage capacity of 450 mio. m$^3$ and is used mainly for irrigation of the lower Arghandab valley (Goes et al., 2016).

Due to the arid climate, natural vegetation is very scarce and mostly limited to seasonal grassland throughout the entire HRB. Irrigated agriculture is by far the largest consumer of water, accounting for 98 % of all abstractions (Goes et al., 2016). Except for a few recent irrigation projects in the LARB and LHRB, irrigation relies on traditional methods with irrigation canals and is thus largely confined to the valley floors along the main river channels (Figure 1b). While the irrigated area in the LARB remained somewhat stable at around 370 km$^2$ (~ 0.3 % of the total HRB) over the last decades, satellite imagery (Landsat 7, ETM+) shows that the total irrigated area in the LHRB more than doubled from < 800 km$^2$ (0.8 %) in the late 1970s to 1650 km$^2$ (1.6 %) in 2011 (Figure 2). More than 200 km$^2$ of the increase in the irrigated area is due to the conversion of seasonal grasslands to high-water-requirement poppy cultivation since the 1990s (Hajihosseini et al., 2019). By 2006 around 690 km$^2$ in the HRB were used for poppy cultivation (UNODC, 2006). In 2011, the main crops in the HRB were wheat (~47 %), poppy (~ 32 – 37 %), maize and beans (~16 %), with orchards in some areas (~1–4 %), and large areas of opium poppy, mostly grown in the traditionally irrigated area (Wardlaw et al., 2013).

## 3 Climatological and hydrological data

The HRB is characterized by poor coverage of reliable historic in-situ observations of hydro-climatic variables, particularly in the upper parts of the basin where most of the water in the HRB originates from. Analysis of Hajihosseini et al. (2016) indicated that the spatio-temporal variation of the interpolated historical precipitation and temperature in the gridded

Climatic Research Unit (CRU) dataset was largely consistent with available ground observations for Afghanistan. Therefore, we here used daily precipitation and temperature estimates for the 1970–2006 study period (Figure 1a), downscaled from the monthly CRU TS 3.10 dataset (Harris et al. 2014), based on the dGen algorithm (Geng et al., 1986) that was previously also applied in other studies (e.g. Schuol and Abbaspour, 2006; Schuol et al., 2008; Hajihosseini et al., 2016). The data were available from www.2w2e.com (Ashraf Vaghefi et al., 2017) at a spatial resolution of $0.5° \times 0.5°$.

Daily streamflow observations for the 1970-1979 period are available from the US Geological Survey (waterdata.usgs.gov) at six gauging stations throughout the HRB (IDs 1-2, 4-7; Figure 1; Table 1). Note, that there were observations available from individual gauging stations at the inlets upstream of the Kajakai Dam (ID1 – $UHRB_U$) and Dahla Dam Reservoirs (ID4 - $UARB_U$)  as well as at the corresponding outlets downstream of the dams (ID2 – $UHRB_D$; ID5 – $UARB_D$). In addition, monthly flow observations for the 1970-2006 period were available at the inflow to the Sistan Plain (ID8 – SISP).

## 4 Methods

The analysis of the characteristics and pattern of hydrological droughts in the HRB over the recent decades in this study required a two-step approach. In a first step, the observed streamflow time series (1970-1979; Table 1) had to be extended to cover the full 1970-2006 study period, using a hydrological model. In a second step, the modelled streamflow estimates for the 1970-2006 period at eight locations in the HRB were used to analyse hydrological droughts.

### 4.1 Hydrological model

We used a distributed implementation of a process-based hydrological model, based on the general concept of the FLEX model-family (e.g. Fenicia et al., 2008; Gharari et al., 2014; Bouaziz et al., 2018) to generate estimates of daily discharge from the sub-basins $UHRB_U$ (ID1), CHRB (ID3), $UARB_U$ (ID4), LARB (ID6), LHRB (ID7) and SISP (ID8). In addition, a simple reservoir routing scheme was used to estimate outflow from the two reservoirs (ID9-10), located at $UHRB_D$ (ID2) and

$UARB_D$ (ID5). The distributed implementation of this model was chosen as the general model set-up was previously successfully applied in climatically similar regions (e.g. Gao et al., 2014, 2017) but also in many other settings worldwide (e.g. Fenicia et al., 2006; Kavetski et al., 2011; Nijzink et al., 2018; Hulsman et al., 2020). In general, the FLEX modelling concept applied here is underlain by a philosophy of model customization and rigorous testing to ensure the implementation of suitable model formulations and the associated more reliable model outputs in different environments (e.g. Fenicia et al.,

2011).

### 4.1.1 Model structure at grid cell scale

The core of the model is five storage components (Figure 3) that are linked by fluxes and that conceptually represent snow storage $S_{sn}$ [mm], interception storage $S_i$ [mm], storage in the unsaturated root-zone $S_u$ [mm], a fast responding component $S_f$ [mm] that generates preferential and overland flow, and a slow responding groundwater storage $S_s$ [mm]. A lag function represents the lag time between storm and flood peak. The snow module is based on a simple degree-day method that has been effectively applied in many conceptual models (e.g. Parajka and Blöschl, 2008; Konz and Seibert, 2010; Gao et al., 2017; Nijzink et al., 2018; Mostbauer et al., 2018). When the average daily temperature is below a threshold temperature $T_t$ [$^oC$], precipitation enters the system as snowfall $P_s$ [mm d$^{-1}$] and is stored in $S_{sn}$. When there is snow cover and the temperature exceeds $T_t$, snow melt $M$ [mm d$^{-1}$], specified by a melt factor $F_m$ [mm $^oC^{-1}$ d$^{-1}$], sets in from $S_{sn}$. Precipitation falling as rain $P_r$ [mm d$^{-1}$] first enters the interception reservoir $S_i$, specified by an interception capacity $I_{max}$ [mm]. Water evaporates as interception evaporation $E_i$ [mm d$^{-1}$] from $S_i$ at potential rates $E_p$ [mm d$^{-1}$], while water in $S_i$ that exceeds the storage capacity $I_{max}$ reaches the soil as throughfall $P_{tf}$ [mm d$^{-1}$]. The total effective precipitation $P_e$ [mm d$^{-1}$] infiltrating into the unsaturated soil root-zone $S_u$ at any given time step is then the sum of $P_{tf}$ and $M$ (Gao et al., 2014). Water in the unsaturated reservoir $S_u$ is, depending on the storage capacity $S_{umax}$ [mm], either stored and eventually released by plant transpiration $E_T$ [mm d$^{-1}$], or directly released as groundwater recharge $R_s$ [mm d$^{-1}$] or preferential flow $R_f$ [mm d$^{-1}$]. The response reservoirs $S_f$ and $S_s$ represent a fast responding storage component and a slower responding groundwater component, respectively, that both drain water to the river according to their associated time scales $k_f$ [d] and $k_s$ [d], so that the total flow can be expressed as $Q=Q_f+Q_s$ [mm d$^{-1}$]. All relevant model equations are provided in Table S1 in the supplementary material.

### 4.1.2 Reservoir routing

Large reservoirs such as the Kajakai (ID9) and Dahla (ID10) Dam reservoirs in the HRB, can considerably alter downstream flow regimes (Haddeland et al., 2014; Wada et al., 2017). This has recently received growing attention and a number of studies have suggested methods to quantify reservoir outflow where reservoir operation rules are largely unknown (e.g. Coerver et al., 2018; Yassin et al., 2019). Here, the effects of the reservoirs were estimated with a simple water accounting scheme based on elevation-storage and elevation-area relationships provided in a study by Vining and Vecchia, (2007) and similar to previous work (e.g. Hanasaki et al., 2006; Wisser et al., 2010):

$$\frac{dS_r(t)}{dt} = Q_{in}(t) - Q_{out}(t) + P(t) - E_p(t), \tag{1}$$

Where $S_r$ is the reservoir storage, $P$ and $E_p$ are precipitation and potential evaporation over the surface area of the reservoir at the end of the previous time step, respectively. $Q_{in}$ is the inflow to the reservoir, $Q_{out}$ the outflow from the reservoir. Here, the inflows $Q_{in}$ to the two reservoirs were estimated by the hydrological models of the respective upstream sub-basins UHRB$_U$ (ID1) and UARB$_U$ (ID4). Due to the lack of more detailed data, $Q_{out}$ was in this study estimated based on empirical

storage-outflow relationships that relate modeled reservoir storage $S_r$ (Eq.1) and $Q_{in}$ to observations of $Q_{out}$, i.e. $Q_{ID2}$ and $Q_{ID5}$. We decided to develop separate linear relationships for high- and low-flow seasons, i.e. January to June and July to December, respectively as preliminary analysis suggested that these were more robust than non- or piecewise-linear relationships for the entire year, as used elsewhere (e.g. Yassin et al., 2019):

$$Q_{out} = \begin{cases} a_h S_{r,t-1} + b_h Q_{in,t} + c_h & \rightarrow high\ flow\ season \\ a_l S_{r,t-1} + c_l & \rightarrow low\ flow\ season \end{cases}, \tag{2}$$

Where $a$ [d$^{-1}$], $b$ [-], $c$ [mm d$^{-1}$] are coefficients and the subscripts h and l indicate high and low flow seasons, respectively. Note, that $Q_{in}$ becomes negligible in the low flow season and the relationship collapses to a simple linear regression. Also note, that it is plausible to assume that reservoir operation is more careful during drier years than in wetter years and may have changed over the study period. Due to the lack of sufficient data, we here developed only one low flow and one high flow relationship for each reservoir over the entire study period.

### 4.1.3 Model implementation at (sub-)basin scale

The model was implemented in a distributed way and the flows aggregated to the (sub-)basin scale. To limit the computational requirements, the meteorological input data, available at a spatial resolution of 0.5° x 0.5°, were averaged to run the model at a grid cell size of 1° x 1° (Figure 1). The snow ($S_{sn}$), interception ($S_i$) and unsaturated ($S_u$) reservoirs in each model grid cell were further stratified into 500m elevation bands to account for elevation-dependent snow dynamics and the associated differences in liquid water input to the system. The combined groundwater recharge $R_s$ and the combined preferential drainage $R_f$ from all elevation zones in each model grid cell was then computed as the weighted average from all individual elevation zones, based on the areal proportion of each elevation zone (cf. Fenicia et al., 2008; Euser et al., 2015). The flow $Q_i$ generated in each of the N grid cells of a (sub-)basin j at any time step t was subsequently routed to the (sub-) basin outlet in a convolution operation with triangular lag functions (e.g. Fenicia et al., 2011) based on lag times $\tau_i$, proportional to the mean flow distances from the individual i cells to the outlet. In addition, irrigation demand $I_D$ [mm d$^{-1}$] for agriculture was accounted for by direct river water abstractions. The aggregated flow at the outlet of a (sub-)basin was then the sum of all flows routed to the outlet minus irrigation demand, i.e. $Q_{ID_j} = \sum_{i=1}^{N} \left( Q_{i,j} * h(\tau_{i,j}) \right) - I_{D,j}$ of that specific (sub-)basin j, i.e. ID1-7. At each time step, irrigation water $I_{D,j}$ was then re-applied as input to $S_{u,i}$ in grid cells i of the corresponding subbasin j that featured agricultural use. Largely being an unregulated irrigation canal system and due to the lack of more detailed information, estimates of $I_{D,j}$ were here based on crop coefficients $K_c$, potential evaporation $E_p$ and effective precipitation $P_e$ for each day k, as well as the agriculturally used area in each year l (Allen et al., 1998), according to $I_D = K_c \left( E_p - P_e \right)_{k,l} A_l$

As a baseline, crop coefficients and the agriculturally used area were estimated based on crop pattern reported by Wardlaw et al., (2013). In that report, the irrigated areas were estimated using satellite imagery from 2010/2011. To account for land-use

change over the 1970-2006 study period, the estimates were adjusted to changes in the agricultural area as extracted from available satellite imagery in 1977, 1988, and 1998.

The outflow of sub-basins UHRB$_U$, i.e.Q$_{ID1}$, and UARB$_U$, i.e. Q$_{ID4}$, were used as inflow to the Kajakai Dam (ID9) and Dahla Dam (ID10) reservoirs, respectively. The resulting estimates of reservoir outflows, i.e. Q$_{ID2}$ and Q$_{ID5}$ (section 4.1.2) were then used as inflows into LHRB (ID7) and LARB (ID6), respectively. In addition, LHRB (ID7) received the outflows Q$_{ID3}$ and Q$_{ID6}$, while LHRB (ID7) outflow Q$_{ID7}$ subsequently drained into SISP (ID8).

The historical absence of significant snow cover in the sub-basins ID2 and ID5-8 allowed us to omit the snow component and the related parameters from the model in these sub-basins (Figure 3) and to limit the adverse effects of equifinality (Beven, 2001). Furthermore, as agriculture is largely confined to the sub-basins LARB (ID6) and LHRB (ID7), the redistribution of river water for irrigation was only implemented in these two sub-basins. Similarly, an additional parameter k$_L$ was used to account forlosses between ID7 (LHRB) and ID8 (SISP). The above differences resulted in two slightly different implementations of the model in the uplands and the downstream regions of the HRB, respectively, and hereafter referred to as Model-1 and Model-2 (Figure 3). Similar implementations of this model type have in the past proven successful in a range of different environments (e.g. Prenner et al., 2018; Hulsman et al., 2020).

### 4.1.4 Model calibration and post-calibration evaluation

The models were run on a daily time step in all sub-basins for the entire 1970-2006 period. However, in the absence of suitable data, the models could not be calibrated for all sub-basins and over the entire period. Rather, only the models of the five sub-basin outlets UHRB$_U$ (ID1), UARB$_U$ (ID4), LARB (ID6), LHRB (ID7) and SISP (ID8) (Table 1; Figure 3) were individually calibrated for the 1970-1975 period to time series of observed flow. Note, that all model grid cells in a given sub-basin were run with the same parameter sets but with spatially distributed hydro-climatic forcing (e.g. Ajami et al., 2004; Euser et al., 2015). To limit the effects of equfinality (Beven, 2001) and to ensure robust model implementation (Euser et al., 2013; Hrachowitz and Clark, 2017), we adopted a multi-objective (Gupta et al., 1998) calibration approach, simultaneously using the Nash-Sutcliffe Efficiency (Nash and Sucliffe, 1970) of flows (E$_{NS,Q}$) and of the logarithm of flows (E$_{NS,log(Q)}$) as objective functions. The 10 (UHRB$_U$, UARB$_U$) and 8 (LARB, LHRB, SISP) free calibration parameters, respectively, in the individual models were sampled in $10^6$ realizations from uniform prior distributions following a Monte Carlo strategy. The model parameters together with their prior and posterior distributions are given Table 2. To account for trade-offs in the multi-objective calibration and uncertainties in the modelling process, we kept all parameter sets that fall into the area spanned by the pareto-optimal set of solutions as feasible (e.g. Fenicia et al., 2007; Gharari et al., 2013). For brevity, we will hereafter refer to the solution with the minimum Euclidean distance D$_E$ as the "best" solution (Hrachowitz et al., 2014):

$$D_E = \sqrt{\left(1 - E_{NS,Q}\right)^2 + \left(1 - E_{NS,log(Q)}\right)^2} \tag{3}$$

Model uncertainty intervals were constructed from the parameter sets that were retained as feasible using $D_E$ as informal likelihood measure to weight each solution (cf. Freer et al., 1996).

In addition, storage-outflow relationships for the reservoirs (ID9-10; Eq.2) to estimate water release from the associated sub-basins downstream of the reservoirs $UHRB_D$ (ID2) and $UARB_D$ (ID5) were established as ordinary least squares estimates based on inflows from the calibrated upstream sub-basins ($UHRB_U$, ID1; $UARB_U$, ID4), Equation 1 and observations of reservoir water release in the 1970-1975 period. The parameter ranges for all solutions retained as feasible for all calibrated hydrological models and both reservoir routing schemes are given in Table 2. Note that due to physiographic similarity, the uncalibrated model for CHRB (ID3) was run with the same parameter sets as $UHRB_U$ (ID1).

The robustness of the calibrated model and its ability to reproduce the time series of daily flow with respect to $E_{NS,Q}$ and $E_{NS,\log(Q)}$ in the four calibration sub-basins as well as downstream of the reservoirs was evaluated for the independent 1976-1979 test period, hereafter referred to as "validation period". In addition, the model output was evaluated against the monthly time series of flow at SISP (ID8; Table 1; Figure 1) for the entire 1976-2006 study period.

## 4.2 Drought indices

Three previously developed drought indices, based on the general concept of standardized deficits (e.g. Moravec et al., 2019), were used here to isolate the individual influences of different factors on hydrological drought in the HRB. The role of climatic variability and thus meteorological drought was quantified with the Standard Precipitation Index (SPI) as introduced by McKee et al. (1993), which gives information about deficits in atmospheric water supply, and with the Standardized Precipitation Evapotranspiration Index (SPEI; Vicente-Serrano et al., 2010), which describes the interaction of precipitation and energy supply as moisture deficit $D_i = \sum_{i=1}^{T}(P_i - E_{p,i})$ and thus the additional role of atmospheric water demand. In contrast, hydrological drought was quantified with the Streamflow Drought Index (SDI; Nalbantis and Tsakiris, 2009). Differences between SPI and SPEI on the one hand and SDI on the other hand were subsequently used to analyse for potential effects of anthropogenic influences, such as irrigation water abstraction. In a parametric approach, two-parameter Gamma distributions functions were here fitted to precipitation P and flow Q and then mapped to standard normal distributions using equal probability transformations (Edwards & McKee, 1997) to estimate the dimensionless drought indices SPI and SDI, respectively (e.g. Lloyd-Hughes and Saunders, 2002; Nalbantis and Tsakiris, 2009; Mishra et al., 2018), whereas generalized extreme value (GEV) distributions were fitted to moisture deficit D to estimate SPEI for each sub-basin (Stagge et al., 2015). The goodness of fit of two-parameter gamma distributions for SPI, SDI, as well as for the GEV distribution for SPEI is provided in Figures S1-S3 in the Supplementary Material. The drought indices can be computed over different time-scales, thus leading to differences in the accumulation of deficits for the corresponding variables (e.g. McKee et al., 1993; Van Loon and Laaha, 2015). Here the drought indices were computed for each month using a time scale of the 12 preceding months as accumulation periods as these were previously found to be the most balanced time scale that gives a balance between short term and long term effects (e.g. Raziei et al., 2009; Gocic et al., 2013; Spinoni et al., 2014). All normalization was carried out relative to the full 1970-2006 study period. Droughts and their

associated occurrence probabilities were classified according to the scheme suggested by McKee et al. (1993) as shown in Table 3. Since the drought indices are standardized, the same drought category thresholds were used here for all three of them.

The three drought indices were in the following used to analyse different drought characteristics. It was investigated if drought frequency, duration, severity and intensity exhibit systematic shifts over time or changes in their longitudinal

propagation from upstream to downstream over the 37 year study period. Drought frequency $D_F$ [months yr$^{-1}$] was here defined as the average number of months per year over a specific period in which the respective drought index, i.e. SPI, SPEI or SDI, had a value < -1 (Table 3). Drought duration $D_D$ [months] was defined as the period of consecutive months with drought indices continuously < -1. Drought severity is defined as the total deficit $D_{tot}$ [-] of SPI, SPEI or SDI, respectively, accumulated during all individual continuous drought periods over a specified period and, to allow

comparability, normalized by the total number of months N in the time period considered, i.e. $D_S = D_{tot}/N$ [month$^{-1}$]. Drought intensity is expressed as the ratio $D_I = D_{tot}/D_D$ [month$^{-1}$] (Huang et al., 2016).

## 5 Results and discussion

### 5.1 Model performance

The hydrological models captured the magnitudes and dynamics of daily flow relatively well when compared to observations

available for both, the sub-basins upstream of the reservoirs, i.e. UHRB$_U$ (ID1; Figure 4a) and UARB$_U$ (ID4) as well as for those further downstream, i.e. LARB (ID6) and LHRB (ID7; Figure 4c). For the calibration period the "best" solutions exceeded $E_{NS,Q} > 0.70$ and $E_{NS,log(Q)} > 0.75$ for all five calibrated sub-basins (Table 4). Similar values were found for the validation period with $E_{NS,Q} > 0.70$ and $E_{NS,log(Q)} > 0.75$. The empirical relationships to route flows through the reservoirs during high and low flow periods (Eq.2) were characterized by $R^2 = 0.80$ and 0.57, respectively for the Kajakai Dam

Reservoir (ID9) and $R^2 = 0.92$ and 0.76, respectively for the Dahla Dam Reservoir (ID10). Although the storage-discharge relationships are statistically significant (p < 0.001), the effect size for low flow periods remains modest. However, a preliminary sensitivity analysis, based on 100 low flow time series of reservoir outflows, sampled from the 5/95$^{th}$ confidence intervals of the low flow storage – discharge relationships suggests that this uncertainty in the relationships has only very limited absolute effects on downstream outputs (Figure S4 in the Supplementary Material). Overall, the resulting flows at

UHRB$_D$ (ID2) and UARB$_D$ (ID5) could be reproduced with $E_{NS,Q} > 0.79$ and $E_{NS,log(Q)} > 0.81$ for the calibration period and comparable performances during the validation period (Figure 4b, Table 4). The ability of these models to reproduce flow in the upstream regions resulted in a robust representation of flow in the downstream Sistan Plain (SISP; ID8) for the entire validation period 1976-2006 without further calibration (Figure 4d, Table 4). Hydrographs of sub-basins not shown in Figure 4 are provided in Figure S5 in the Supplementary Material.

In general, the estimated water release from the reservoirs results in overall model outputs in all downstream basins are widely consistent with the observed daily river flow, which at station SISP (ID8) is even true for the entire 37-year study

period. In spite of all other sources of uncertainty throughout the modelling process, this can be seen as an indication of the plausibility of the modelled reservoir outflow.

It could be observed that annual peak flows in spring are mostly generated by a combination of snowmelt from the high-
elevation parts of the HRB, i.e. in sub-basins ID2, 3 and 4, and additional, relatively high-intensity rainfall events (Figure 4). The filling of the two reservoirs attenuates downstream flows, including the annual peaks, throughout springand into early summer. In turn, the gradual release of water from the reservoirs sustains downstream summer and autumn flows, almost doubling long-term average low flow rates as compared to natural flow conditions (Figures 4, 5), to meet irrigation demand in the downstream Helmand Valley and to satisfy flow requirements of the Sistan River in Iran under the Iranian-Afghan
Helmand River Water Treaty (1973).

Furthermore, the models adequately reproduced the losing character of the downstream sub-basins, including LHRB (ID7) and SISP (ID8). Thus, in this highly water-limited environment, these sub-basins do not generate relevant volumes of flow. Rather, most of the precipitation and, in addition, significant volumes of water entering LHRB (ID7) and eventually SISP (ID8) as flow from upstream, eventually evaporate. Besides this, streamflow draining LHRB (ID7) and crossing a hyper-arid
desert region is reduced by about 60% before reaching SISP (ID8), as specified by the calibrated loss factor $K_L$. These streamflow reductions cannot be explained by deep infiltration losses and soil evaporation alone in this essentially vegetation-free environment. There is another even much more plausible source of these observed and modelled flow reductions: when the Helmand River reaches Iran, it bifurcates just upstream the gauge at SISP (ID8) into the Sistan river (SISP, ID8), which drains into the Hamun wetlands, and the completely ungauged Common Parian River, which follows the
border between Iran and Afghanistan. The magnitudes of flow diversion are undocumented and merely Burger (2005) in a study of the Helmand River of Afghanistan and Iran loosely mentioned potential uncertainties arising from this diversion into the Common Parian River. Therefore the lumped loss factor ($K_L$) combines the effects of deep infiltration (e.g. Schaller and Fan, 2009; Bouaziz et al., 2018; Condon et al., 2020), evaporation and particularly the proportion of water which is diverted into the Common Parian River.

Overall, following a multi-objective calibration strategy, i.e. simultaneously using $E_{NS,Q}$ and $E_{NS,\log(Q)}$ as calibration objectives to ensure good representation of both, high- and low flows, our model performances with respect to daily flow in all sub-basins (Table 4, Figure 4), exceed those of the studies of Hajihosseini et al. (2016) but also those of Hajihosseini et al. (2019) who assessed the monthly flow with the SWAT model in the Upper and Lower Helamand basins, respectively.

## 5.2 Temporal pattern of drought

### 5.2.1 SPI

Multiple meteorological drought events in terms of SPI occurred in the HRB throughout the 1970-2006 study period (Figure 6a). An average mean drought frequency across all sub-basins of the HRB of $D_{F,SPI}$ = 2.5 months year$^{-1}$ characterized the 1970-1979 decade. This is higher than in the subsequent two decades during which $D_{F,SPI}$ reached 0.5 and 0.3 months year$^{-1}$, respectively. The last part of the study period, 2000-2006, experienced more precipitation deficits again, resulting in frequent drought spells with $D_{F,SPI}$ = 5.4 months year$^{-1}$. A similar pattern was found for drought duration. While the two decades in

the middle of the study period experienced mean drought durations across all sub-basins between $D_{D,SPI}$ = 1.2 and 1.6 months, much longer droughts occurred in the first and last decades with $D_{D,SPI}$ = 15.1 and 21.1 months, respectively (Figure 6a). Reflecting the above, the mean drought severity and intensity were also more pronounced at the beginning and towards the end of the study period, with the lowest mean $D_{S,SPI}$ = -0.7 and $D_{I,SPI}$ = -1.6 month$^{-1}$, respectively, in the 2000-2006 period, as compared to the highest $D_{S,SPI}$ = -0.1 and $D_{I,SPI}$ = -0.8 month$^{-1}$ in the wetter period between 1980 and 1999.

Notwithstanding the fluctuating pattern in these drought descriptors over the study period, pairwise comparisons of the decadal distributions of basin-average annual SPI values using Wilcoxon rank-sum tests indicated that there is no significant difference between any of the decadal SPI distributions  (p > 0.05), as also shown in Figure 7a. Correspondingly, no temporal trends in the time series of annual SPI could be detected based on Mann-Kendall tests (Kendall, 1975) for the HRB or any sub-basin therein (p > 0.05; Figure 7b). The outputs of the drought analysis with a discretization of the study period

into 2 20-year periods resulted in equivalent interpretations (Supplementary Material Figure S6a): in spite of slightly more pronounced $D_{F,SPI}$, $D_{D,SPI}$ and $D_{S,SPI}$ in the 1990-2006 period, the differences to the 1970-1989 period are statistically not significant (p > 0.05).

Although these results support the findings of Miyan et al., (2015) who reported that Afghanistan experienced unusual droughts from 1995 onwards until the heavy snow falling in the 2002-2003 winter season, precipitation and the associated

meteorological drought did, in spite of decadal fluctuations, not experience a systematic change in the HRB over the four study decades.

### 5.2.2 SPEI

The temporal pattern of drought in terms of SPEI, reflecting the combined effects of precipitation water supply and atmospheric water demand, similarly indicates the occurrence of multiple periods of severe drought in all sub-basins

throughout the HRB during the 1970-2006 study period (Figure 6b). The temporal fluctuations in SPEI broadly correspond with those in SPI, suggesting that most drought events are largely controlled by water supply and thus precipitation deficits rather than by increased atmospheric water demand in this arid region. More specifically, mean drought frequency across all sub-basins decreased from $D_{F,SPEI}$ = 2.8 months year$^{-1}$ in the 1970-1979 decade to around 0.2 months year$^{-1}$ in the following two decades. In the last decade of the study period, however, a pronounced increase in drought frequency to $D_{F,SPEI}$ = 6.1

months year$^{-1}$ was observed (Figure 6b). While individual drought events had average durations between $D_{D,SPEI} = 0.8$ and 2.4 months in the two middle decades across all sub-basins, this was substantially higher $D_{D,SPEI} = 9.9$ months in the first decade and even increased to 15.8 months in the extreme drought of the 2000-2006 decade. Drought severity and intensity remained at relatively modest levels not falling below $D_{S,SPEI} = -0.1$ and $D_{I,SPEI} = -0.8$ month$^{-1}$, respectively, in the 1980-1999 period. In contrast, the first and last decade were characterized by much more pronounced severity and intensity, with the lowest mean $D_{S,SPEI} = -0.8$ and $D_{I,SPEI} = -1.5$ month$^{-1}$, respectively, occurring during the 2000-2006 period. Similar to SPI, Wilcoxon rank-sum tests showed that there is mostly no systematic and significant difference between the decadal distributions of basin-average SPEI ($p > 0.05$), with the exception of the 2000-2006 decade, during which SPEI is significantly lower than during the 1990-1999 decade for most sub-basins ($p \leq 0.05$), as shown in Figure 7c. The temporal sequence of a slight SPEI increase during the first three decades followed by a sharp decrease during the multi-year drought in 2000-2006 likewise illustrates that there is no systematic trend in the time series of SPEI in the HRB or any sub-basin therein over the study period ($p > 0.05$; Figure 7d). The results of the same analysis over two 20-year periods similarly suggest that the 1990-2006 period experienced was slightly more drought affected with somewhat more frequent, longer and more severe droughts as compared to the 1970-1989 period, yet the overall differences are statistically not significant ($p > 0.05$; Supplementary Material Figure S6b).

**5.2.3 SDI**

Streamflow drought, as specified by SDI, was quantified based on streamflow estimates as obtained from the best available model solution for each of the eight sub-basins. It could be observed that SDI largely follows the temporal pattern in SPI and SPEI (Figure 6c), respectively, with a relatively low lag time of $\leq 1$ month in all sub-basins throughout the HRB, as suggested by a cross-correlation analysis between time series of monthly SPI, SPEI and SDI in the individual sub-basins ($r = 0.66 - 0.91$; $p < 0.05$; not shown). However, it can also be observed that, overall, SDI drought events are less pronounced than SPI and SPEI droughts occurring at around the same time. More specifically it was found that the mean drought frequency across all sub-basins fluctuated between $D_{F,SDI} = 0.1$ and 2.0 months year$^{-1}$ in the first three decades of the study period. In the 2000-2006 decade, it experienced a marked increase to $D_{F,SDI} = 6.7$ months year$^{-1}$. Similarly, mean drought duration was with $D_{D,SDI} = 21.0$ months highest in that decade. In the other these decades the mean $D_{D,SDI}$ did not exceed 11.3 months. Closely reflecting the pattern of SPI and SPEI, mean drought severity and intensity across all sub-basins were most pronounced in the first and last decades with both $D_{S,SDI}$ and $D_{I,SDI}$ reaching minimum values of -0.9 and -1.5 month$^{-1}$, respectively, in the 2000-2006 period. During the wetter decades in-between $D_{S,SDI}$ and $D_{I,SDI}$ did not decrease below values of -0.1 and -0.9 month$^{-1}$, respectively.

Following a pairwise comparison of all decadal basin-average SDI distributions, the slight, yet statistically insignificant increase of the decadal SPI and SPEI distributions ($p > 0.05$; Figures 7a,c) from 1970-1999 could not be observed in SDI, which remained rather stable during the first three decades (Figure 7e). In contrast, the decrease of SPEI in the last decade is reflected in correspondingly lower basin-average SDI in the 2000-2006 period ($p \leq 0.05$; Figure 7e) (Li et al., 2019;

Noorisameleh et al., 2019). However, the time series of basin-average SDI did not exhibit a significant trend ($p > 0.05$; Figure 7f). In contrast, comparison of the two 20-year periods (1970-1989 and 1990-2006) suggests higher drought

frequency, longer duration and more pronounced severity and intensity, respectively, as compared to the 1970-1989 period. Based on a Wilcoxon rank-sum test, there is a systematic and significant difference between the basin-average SDI distributions for the two 20-year periods ($p > 0.05$, Supplementary Material Figure S6c), indicating a shift from mild to severe hydrological drought in the study area.

### 5.3 Spatial pattern, synchronicity and propagation of drought

### 5.3.1 SPI

In most years of the study period meteorological drought, as specified by SPI, exhibits considerable spatial coherence and synchronicity throughout the HRB (Figure 6a). In other words, at any given time, the entire HRB experiences similar relative precipitation deficits (or surpluses), with a median $r = 0.97$ ($p < 0.05$) as obtained from a Spearman rank correlation between the time series of SPI across all sub-basins. Regional differences in SPI remain limited to parts in the central HRB, i.e.

CHRB (ID3) and LARB (ID6, Figure 6a). In contrast to the remainder of the HRB, these two sub-basins are characterized by multiple periods that are, in relative terms, more humid, such as in 1974 or 1982, but also by periods that are, in comparison, considerably drier, such as 1987 or 1994. The elevated degree of spatial coherence and synchronicity in SPI on the scale of the HRB is further illustrated by the comparison of the upstream and downstream decadal SPI distributions (Figure 8a). No significant differences ($p > 0.05$) between the SPI distribution of the six most upstream sub-basins (ID1-ID6) and the SPI

distribution of the two most downstream sub-basins, LHRB (ID7) and SISP (ID8), could be found in any of the four decades during the study period. To provide some more explicit spatial context, the spatial distribution of SPI at the resolution of the individual model grid cells for four selected years is shown in Figure 9a-d. Compared to the SPI aggregated at the scales of the individual sub-basins (Figure 6a), this more detailed picture corroborates the level of large-scale spatial coherence, in spite of somewhat increased local variations in SPI (Figure 9a-d). A rather rare exception is the year 1987, which was

characterized by a substantial North-South gradient in SPI spatial variations and whose extent is largely masked by the aggregation of SPI to the sub-basin scale in Figure 6a.

### 5.3.2 SPEI

While SPEI is widely coherent (median $r = 0.94$, $p < 0.05$) and spatially broadly follows the pattern of SPI throughout large parts of the HRB, it can also be observed that inter-annual differences in atmospheric water demand, here estimated based on

$E_P$, lead to modest, yet contrasting effects (Figure 6b). For some sub-basins and time periods characterized by comparably cool temperatures, water deficits are attenuated and SPEI thus remains higher than SPI (e.g. UARB$_U$-ID4 in 1986 or LARB-ID6 in 1989). For other sub-basins and warmer periods, increased atmospheric water demand reinforces water deficits (e.g. CHRB-ID3 in 1981). As shown in Figure 8a, the distributions of SPEI closely reflect the distributions of SPI in the first

decade of the study period. In the following 1980-1989 decade as well as in the 2000-2006 decade SPEI is lower than SPI,

potentially indicating the role of $E_P$ in intensifying water deficits in these periods. In contrast, the opposite effect can be observed during the 1990-1999 decade, where rather low $E_P$ had a moderating effect, leading to higher values of SPEI than SPI. Although these effects occur across the entire HRB, water deficits in terms of SPEI are considerably more sensitive to fluctuations in atmospheric water demand and the differences between SPEI and SPI are thus more pronounced in the downstream parts of the HRB (Figure 8a). In particular, SPEI in the hyper-arid SISP (ID8; Figure 6b) is characterized by a

low degree of coherence and synchronicity compared to upstream SPEI, exhibiting both, markedly more severe water deficits (e.g. 1973, 1984 or 2003) but also more pronounced water surpluses (e.g. 1986, 1996 or 2005). Notwithstanding these varying effects of $E_P$ on water deficits and thus on the differences between SPEI and SPI, no systematic temporal trend of $E_P$ reinforcing/moderating water deficits could be detected. However, note that the applicability of SPEI in arid areas such as the study region may be limited (Pei et al., 2020). In such environments such fluctuations in $E_P$ will have a limited effect

on $E_A$ and thus on water deficits as the systems are, by definition, water limited rather than energy limited. Changes in $E_P$ will therefore be less relevant for the intensification/moderation of drought in such arid regions than changes in precipitation.

### 5.3.3 SDI

Hydrological drought in terms of SDI during the study period exhibited a lower degree of spatial coherence and synchronicity (Figure 6c) across the HRB. This is reflected by a lower median r = 0.85 (p < 0.05) from pairwise Spearman

Rank correlations between the individual time series of SDI across all sub-basins. The spatially and temporally more heterogeneous mosaic of SDI, however, allows a few insights. The data suggest that both reservoirs, at Kajakai dam and Dahla dam, respectively, have effects on the propagation of hydrological drought. This can be seen in the differences in SDI between the sub-basins upstream (UHRB$_U$-ID1; UARB$_U$-ID4) and the associated sub-basins downstream of the dams (UHRB$_D$-ID2; UARB$_D$-ID5) in Figure 6c. In the early phase of the study period, the reservoirs had some moderating effects

on the propagation of hydrological droughts, most notably for the 1977 (both dams) and 1971 (Dahla dam) droughts. The median SDI in the 1970-1979 decade was ~0.2 higher downstream than upstream of both reservoirs (p < 0.05). However, over the following decades, both reservoirs largely lost their drought attenuating functions and the reservoir at Dahla dam may have even contributed to amplifying the 1999-2002 drought downstream of the dam, with a median SDI over that period being ~0.18 (p < 0.05) lower at the downstream UARB$_D$ (ID5) than the upstream UARB$_U$ (ID4).

While the distribution of SDI broadly follows the distributions of SPI and SPEI in the upstream part of the HRB (ID1-ID6), downstream hydrological drought is characterized by rather distinct dynamics (ID7-ID8; Figure 8a). In contrast to the basin-average time series of SDI (Figure 7f), SDI in the two downstream sub-basins exhibit clear negative trends over the four decades of the study period (p ≤ 0.05; not shown). In addition, the data suggest that for the 1970-1979 decade the median downstream SDI ~ 0.2 is significantly higher (p < 0.05) than SPI, SPEI and upstream SDI, which are all characterized by a

median of about -0.1 (Figure 8a). As also shown by the individual SDI distributions of all sub-basins in Figure 8b, hydrological drought is considerably attenuated and the relative river water deficits reduced compared to upstream parts of

the HRB during that period. However, throughout the following two decades, the downstream moderation of hydrological drought weakens, i.e. the distributions of downstream SDI more closely reflect those of SPI, SPEI and upstream SDI (Figure 8b). This pattern then eventually fully inverts into a downstream drought amplification in the 2000-2006 decade, during which the median downstream SDI = -1.5 is significantly lower ($p < 0.05$) than not only the median upstream SDI = -0.9 but also than SPI and SPEI (Figures 8a,b). This shift from downstream drought moderation to drought amplification can be seen clearly for the four selected years in Figure 9e-h. In spite of dry meteorological conditions throughout the HRB in 1977 (Figure 9a) and severe hydrological drought in the North of the HRB, no drought occurred in the South of the study region (Figure 9e). In 1987, similarly, the increasing precipitation deficits from upstream to downstream (Figure 9b) were buffered and not reflected in the North-South gradient of SDI, indicating the wettest conditions in the most downstream part of the HRB (Figure 9f). The extreme opposite of the above two examples occurred in the last decade of the study period. In both years, 2002 and 2003, respectively, spatially relatively coherent precipitation pattern across the entire HRB (Figure 9c-d) led to severe hydrological drought in the most downstream parts of the HRB in particular at SISP (ID8; Figure 9g-h). This is particularly striking for the rather wet year 2003, in which SDI in the upstream sub-basins reflected the generally wet conditions of that year, while further downstream river water deficits developed, gradually amplifying to severe drought at SISP (ID8). Further analysis of the time series of the difference between upstream (ID1-ID6) and downstream (ID7-8) SDI (i.e. ΔSDI) shows the inversion from a negative to a positive ΔSDI over the 37 years of the study period occurred gradually and, according to a Mann-Kendall test, following a significant trend ($p < 0.05$), while the differences in SPI remain stable over time (Figure 8c) (Ma et al., 2019). This suggests that it may not be implausible to assume that the inversion of downstream hydrological drought moderation in the 1970-1979 decade into drought amplification in the 2000-2006 decade was, at least partly, an effect of systematic, longer-term shifts in the system rather than a short-term, synchronous occurrence of multiple drought-amplifying hydro-meteorological conditions, such as sustained high precipitation deficits and high atmospheric water demand (Van Loon, 2015). Such short-term influences of deficits in hydrological drivers would be likely to manifest themselves in the evolution of ΔSDI characterized by a more erratic temporal pattern.

**5.4 Drought drivers and process attribution**

The above drought indices provide only limited information to identify dominant drivers of droughts. To gain more understanding of the spatio-temporal pattern of hydrological drought and to eventually attribute droughts to physical processes estimate of the absolute magnitudes of multiple modelled hydrological fluxes, as obtained from the best available model solution for each sub-basin, are in the following analysed.

With a long-term mean annual precipitation of ~250 mm $y^{-1}$ in the HRB, the overall magnitudes of streamflow deficits, and thus of hydrological droughts, are clearly dominated by fluctuations in precipitation anomalies (Figure 10a), with a mean absolute anomaly of around ±50mm $y^{-1}$ for the entire HRB or ~ 20% of the long-term mean water balance. In contrast, anomalies in total evaporation $E_A$ (here: $E_A = E_I + E_T + I_D$) exhibit much lower variability in this arid environment, with a mean absolute anomaly of about ±20 mm $y^{-1}$. As water supply is the limiting factor for evaporation, the highest rates of $E_A$

occur in the wettest years (Figure 10b). Conversely, $E_A$ has a proportionally less impact on streamflow in dry years. In general it can be seen that precipitation anomalies of ~ -50 – -100 mm y$^{-1}$ lead to streamflow anomalies of ~ -20 – -30 mm y$^{-1}$ (Figure 10c). Although SPEI is typically based on potential evaporation $E_P$, arid and thus water-limited environments are rather insensitive to fluctuations in $E_P$ compared to fluctuations in P. In other words, there will be little difference in the partitioning of water fluxes if under the same annual precipitation of e.g. 500 mm yr$^{-1}$, $E_P$ is 1000 or 1500 mm yr$^{-1}$, as in both

cases actual evaporation $E_A$ will be close to (or even exceed) 400 mm yr$^{-1}$ and as therefore most of the available water will be evaporated. In contrast, more water will be evaporated as $E_A$ (even if $E_P$ remains stable) in years when more water is available and thus P is higher. By extension, the effects of evaporation on droughts in arid regions can only be meaningfully assessed by changes in $E_A$. Haung et al., (2017) mentioned that actual evaporation affected strongly the propagation time of meteorological to hydrological drought in the Wei River Basin (WRB), a typical arid and semi-arid region in China.

The modelled data suggest that during drought years, the reservoir at Kajakai Dam released slightly less water (UHRB$_D$-ID2) than it received as an inflow (UHRB$_U$-ID1), as shown in Figure 4. The mean difference between drought period inflow to and outflow from the reservoir remained stable at $\Delta Q$ ~ 0.9 mm y$^{-1}$ throughout the four decades of the study period. This implies that there is no evidence that the reservoir neither moderated nor significantly amplified downstream propagation of streamflow deficits, underlining the very minor role of this reservoir for the drought pattern. In contrast, the modelled flow

estimates for the reservoir at Dahla Dam suggest that this reservoir had some moderation effect on downstream flow deficits and thus drought propagation in the first decade of the study period (Wang et al., 2019). On average, the reservoir outflow (UARB$_D$-ID5) during drought periods in that decade exceeded the inflow (UARB$_U$-ID4) by $\Delta Q$ ~ 1.1 mm y$^{-1}$ (Supplementary Figure S1). However, this difference gradually decreased over time and eventually converged towards zero in the 2000-2006 period. In spite of uncertainties arising from data and the modelling process, this nevertheless indicates the

possibility that the Dahla Dam reservoir has lost its, albeit very minor, drought-moderating function over the study period. This broadly corresponds with the results of Wu et al. (2019) who found that the influence of reservoirs on long-term hydrological drought is limited and may even increase the duration and severity of a drought, whereas shorter hydrological droughts may be shortened and moderated by adequate reservoir operation.

For further analysis the HRB was separated into an upper and a lower basin. The upper basin comprises UHRB$_D$ (ID2),

CHRB (ID3) and LARB (ID3), which together drain into the lower basin, here defined as LHRB (ID7) only and thus for clarity of presentation excluding SISP (ID8). As illustrated by Figure 10, and consistent with the spatial analysis of drought indices in Section 5.3, the general pattern of anomalies corresponds well between the upper and the lower basin, suggesting a considerable level of spatial coherence and drought synchronicity. However, reflecting the evolution of $\Delta$SDI (Figure 8c), a subtle but gradual shift in the difference between streamflow anomalies of the upper and lower basins from, on average, -9.4

mm y$^{-1}$ in the 1970-1979 decade to 5.5 mm y$^{-1}$ in the 2000-2006 period is evident (Figure 10c). Thus, while anomalies were less negative/more positive, therefore indicating proportionally "more" water, in the lower than in the upper basin at the beginning of the study period, the opposite was true at the end of the study period. This entails that in the first decade of the study period streamflow deficits from the upper basin were to some degree attenuated in the lower basin. This effect was

gradually reduced and finally completely inversed in the last decade of the study period. During the 2000-2006 period
streamflow anomalies from the upper basin were systematically amplified in the lower basin. The absence of a similar
systematic shift in the difference of precipitation anomalies between the upper and the lower basin (Figure 10a) strongly
suggests alternative reasons for the above effects. Mianabadi et al. (2020) similarly indicated that in the lower Helmand
River Basin water availability issues cannot be attributed to the changes in precipitation in the downstream area itself.

The analysis of the relative contributions of different water fluxes from the upper and lower basins, respectively, as well as
their evolution over time as estimated from the models allowed some more detailed insights into these patterns. The
combined water balance of all three sub-basins of the upper basin for the 1970-1979 period (Figure 11a) shows that of the
mean annual precipitation $P \sim 202$ mm $y^{-1}$ of the upper basin, 28% drained away as streamflow ($Q \sim 56$ mm $y^{-1}$) and the
remainder of 72% was released as combined evaporative fluxes ($E_A \sim 146$ mm $y^{-1}$). While transpiration ($E_T \sim 130$ mm $y^{-1}$)
and interception evaporation ($E_I \sim 9$ mm $y^{-1}$) played a role throughout the entire upper basin, irrigation demand ($I_D \sim 7$mm $y^{-1}$) was limited to the agriculturally used areas of the LARB (ID6) sub-basin and thus only accounted for $\sim 4\%$ of the water
balance of the upper basin. The flow partitioning of the lower basin for the same period exhibited a considerably different
pattern. It can be seen in Figure 11a that of the available water in the lower basin ($\sim 97$ mm $y^{-1}$), i.e. precipitation over the
LHRB (ID7) sub-basin plus the combined inflow from the upper basin, 51% ($Q \sim 49$ mm $y^{-1}$) is drained as streamflow and
49% are released as evaporative flux ($E_A \sim 48$ mm $y^{-1}$). In comparison to the upper basin, irrigation demand in the lower
basin is with $I_D \sim 13$ mm $y^{-1}$ a substantially larger fraction of the water balance ($\sim 14\%$) than in the upper basin.

During the 2000-2006 period (Figure 11b), the upper basin received slightly less precipitation ($P \sim 179$ mm $y^{-1}$) than in the
1970-1979 period. However, the relative contributions of the different fluxes remained rather stable over time. The fraction
of water drained as streamflow slightly decreased to 25% ($Q \sim 44$ mm $y^{-1}$), while the fraction of evaporative fluxes
correspondingly increased to 75 % ($E_A \sim 135$ mm $y^{-1}$) of the water balance of the upper basin with similar increases for all
three evaporative components (Figure 11b). In contrast, substantial shifts in the flux partitioning can be observed for the
lower basin (Figure 11b). In spite of a reduction of available water to $\sim 71$ mm $y^{-1}$ in the 2000-2006 period, the evaporative
release ($E_A \sim 49$ mm $y^{-1}$) reached the same level as in the 1970-1979 decade. As illustrated by Figure 11b, the high levels of
evaporative release were sustained by significant absolute and relative increases of irrigation demand to $I_D \sim 23$ mm $y^{-1}$ or
32% of the water available in the lower basin (or ~10% of the water balance of the entire HRB). This, in turn, resulted in a
reduction of streamflow to $Q \sim 22$ mm $y^{-1}$, equivalent to a reduction from 51% of the water balance in the 1970-1979 decade
to 31% in the 2000-2006 decade. The increases of $I_D$ and the corresponding decreases in Q are directly related to increases in
the agricultural area over the study period (Figure 2). It is therefore plausible to assume that the inversion of the function of
the lower basin from moderation to amplification of flow deficits and the associated droughts is largely a consequence of
increases in an agriculturally used area which resulted in increases of the related irrigation water demand (AghaKouchak et
al., 2015; Van Loon et al., 2016; Haile et al., 2019). In addition, inefficient irrigation schemes in the study region may lead
to an underestimation of the actual irrigation water use. Therefore higher real-world irrigation water demand would even
further strengthen the results, that the shift form of downstream moderation to the intensification of hydrological drought

over the study period is largely an effect of human intervention. Margariti et al. (2019) found that human activities prolonged drought durations in all the European catchments they analysed. Ma et al., (2019) showed that human inventions are likely to have changed the positive correlation between meteorological and hydrological droughts to negative in the semiarid Heile River Basin, especially during warm and irrigation seasons.

Overall, the magnitudes of flow deficits and the associated hydrological droughts are largely driven by precipitation deficits across the HRB. The two reservoirs in the HRB had a very minor effect on the propagation of flow deficits, with levels not exceeding 0.5% of the water balance of the HRB in the study period. Burger (2005) stated that the extreme drought in the Sistan region between 1999 and 2004 was not caused by the Afghan reservoirs, but rather this period was extremely dry in the whole catchment but in the future, the drought might be worsened due to extend the Kajakai reservoir. In contrast, the increase of agricultural area, mostly in LHRB (ID7), led to an increase of the basin-wide irrigation water demand (i.e. from LARB-ID6 and LHRB-ID7) from ~ 7% to ~ 12% of the water balance of the HRB. While at the scale of the entire HRB this remains of minor relevance for flow deficits, and thus hydrological drought, it led throughout the study period to a continuous and gradual change in the downstream propagation of flow deficits from moderation to amplification. This clearly underlines the argument by Haile et al. (2019), indicating that human activities such as expansion of cultivation, overexploitation of water resources, particularly for irrigation demands have an impact on altering the hydrological processes which are directly linked to drought. Our results further illustrate that flow deficits and droughts in the HRB clearly reflect the dynamic interplay between temporally varying regional differences in hydro-meteorological variables together with subtle and temporally varying effects linked to direct human intervention (Jahanzaib et al., 2020; Jiang et al., 2019; Saeidi et al., 2018, Wan et al., 2017).

**5.5 Uncertainties, unresolved questions and limitations**

All the above results are necessarily conditional on a range of uncertainties and choices made during the modelling process (Beven, 2006; Hrachowitz and Clark, 2017). This is in particular relevant in the HRB, where detailed and reliable data are scarce. It entails further, that although the results of this study are largely consistent with the available data, the data themselves may inaccurately reflect reality. In addition, whereof sufficient quality, the available data may not have sufficient detail to accurately represent the underlying mechanistic processes and/or changes thereof over time in a model. Two major sources of uncertainty, due to the lack of detailed and high-quality data need to be explicitly highlighted for this study. First, the routing of flows through the two reservoirs in the HRB was estimated with a simple empirical relationship (Eq.2) based on data from the 1970-1979 period, under the assumption that this relationship did not change over time. In reality, reservoir operation rules may have changed over the study period, yet this cannot be clarified with the available data. However, even if such changes occurred, their impact is likely limited, as model evaluation at SISP (ID8) showed that adequate model performances were achieved throughout the entire study period (Table 3, Figure 4).

A second not completely resolved issue is the observed and modelled considerable reduction of stream flow between LHRB (ID7) and SISP (ID8). The loss of ~ 60% of streamflow as the river crosses the desert region between Afghanistan and Iran

can plausibly be explained by the combined effects of evaporation, deep infiltration losses and, most importantly, river water diversion into the completely ungauged Common Parian River that bifurcates from the Helamnad River just upstream of SISP (ID8). In the model it was represented by an unspecified loss factor $K_L$. A clearer mechanistic interpretation was not warranted by the available data. Another cause that cannot be completely ruled out is a potentially low quality of the available streamflow data either at LHRB (ID7), at SISP (ID8), or both of them.

We explicitly reiterate here that although this modelling study allowed robust insights into the pattern of drought characteristics, including changes in downstream drought propagation over time, the absolute magnitudes of variables reported herein remain, for the above reasons, highly uncertain. These magnitudes should therefore, under no circumstances and without more detailed data and understanding of the underlying processes, be used for direct policy advice in this arid environment where the transboundary nature of the HRB makes water scarcity a highly sensitive issue.

Clearly, the most reliable way forward to reduce uncertainties in flow estimations in the the HRB is to do more observations and generate data, which can then be confronted with the model. Deficiencies in the model to reproduce these additional data will then, in an iterative process, allow model improvement (e.g. Fenicia et al., 2008; Hulsman et al., 2020). However, at this stage, further model improvement is problematic as the introduction of more complexity in the model will not be warranted by the available data and eventually merely lead to increased equifinality (Beven, 2006). Although beyond the scope of this study, a future comparison of alternative data sets to inform the model and an analysis of the associated potential differences in model results will be highly valuable to ensure reliable interpretations of the HRB and thus to limit uncertainty for better policy advice on water resource management in the HRB.

**6 Conclusions**

In combined data analysis and modelling study in the transboundary Helmand River Basin (HRB) we analysed spatial patterns of drought and changes therein over the 1970-2006 study period, based on the drought indices SPI, SPEI and SDI, as well as on absolute water deficits. The results provide some evidence that:

(1) Drought characteristics varied throughout the study period. In general, the 2000-2006 and partly the 1970-1979 periods were drier than the decades in between. Depending on the drought index, mean drought duration reached $D_D \sim 10 - 20$ months and mean drought intensity $D_I \sim -1.5$ month$^{-1}$ in these drier periods, as compared to $D_D \sim 0 - 2$ months and $D_I \sim -0.1 - -1.0$ in the 1980-1999 period.

(2) The basin-average decadal distributions of the drought indices largely exhibited no statistically significant differences, with the exception of significantly lower SPEI and SDI in 2000-2006 compared to the preceding decades. In addition, no systematic trend over time was detected for any of the basin-average drought indices.

(3) All three drought indices exhibit considerable spatial coherence and synchronicity across the HRB throughout the study period. This indicates that in most cases droughts similarly affect the entire HRB with little regional or local differences.

(4) The overall magnitudes of streamflow drought in the HRB are consistently controlled by precipitation deficits, while the effects of the two reservoirs, as well as water abstraction for irrigation on flow deficits, remain minor during drought years, accounting for only 0.5 % and ~10%, respectively, of the water balance of the HRB.

(5) The downstream parts of the HRB moderated the further propagation of streamflow deficits and the associated droughts in the early decades of the study period. This drought moderation function of the lower basin was gradually and systematically inverted by the end of the study period, when the lower basin eventually amplified the downstream propagation of flow deficits and droughts.

(6) The shift from drought moderation to drought amplification in the lower basin is very likely a consequence of

645 agricultural activity and the associated increased irrigation water demand in spite of being only a minor fraction of the water balance.

Overall the results of this study illustrate that the flow deficits and associated droughts in the HRB clearly reflect the dynamic interplay between temporally varying regional differences in hydro-meteorological variables together with subtle and temporally varying effects linked to direct human intervention.

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

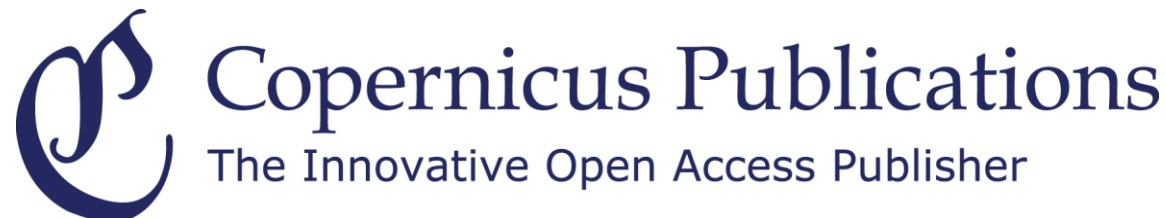






**Table 1**. Summary of sub-basins and reservoirs in the Helmand River Basin. UHRB, CHRB and LHRB denote the Upper,
Central and Lower Helmand River Basins, respectively, UARB and LARB are the Upper and Lower Arghandab River
Basins, respectively and SISP is the Sistan Plain. The subscripts U and D are inflows into the reservoir upstream of the dams
and outflows from the reservoirs downstream of the dams, respectively.

| Location ID | Sub-basin Symbol | Station name | Latitude (°) | Longitude (°) | Average Elevation (m) | Precipitation (mmy$^{-1}$) | Discharge (mmy$^{-1}$) | Aridity Index (-) | Observation Period (daily/monthly) | Reservoir Storage (10$^6$ m$^3$) |
|---|---|---|---|---|---|---|---|---|---|---|
| 1 | UHRB$_U$ | Dehraout – Kajakai Dam inflow | 32.42 | 65.28 | 2865 | 300 | 130 | 0.29 | 1970–1979 (d) | ---- |
| 2 | UHRB$_D$ | Kajakai Dam outflow | 32.19 | 65.06 | 2798 | 300 | 111 | 0.29 | 1970–1979 (d) | ---- |
| 3 | CHRB | - | - | - | 1994 | 204 | - | 0.13 | - | |
| 4 | UARB$_U$ | Upper Arghandab – Dahla Dam inflow | 31.50 | 65.52 | 2830 | 254 | 91 | 0.22 | 1970–1979 (d) | ---- |
| 5 | UARB$_D$ | Dahla Dam outflow | 31.57 | 66.02 | 2776 | 254 | 88 | 0.22 | 1970–1979 (d) | ---- |
| 6 | LARB | Qala-i-Bust | 31.30 | 64.23 | 2509 | 225 | 32 | 0.13 | 1970–1980 (d) | ---- |
| 7 | LHRB | Char Burjak | 30.17 | 62.02 | 2610 | 229 | 43 | 0.16 | 1970–1979 (d) | ---- |
| 8 | SISP | Sistan | 30.49 | 61.46 | 2584 | 250 | 15 | 0.16 | 1970–2006 (m) | ---- |
| 9 | | Kajakai Dam Reservoir | 32.19 | 65.10 | 963 | ---- | ---- | | ---- | 1800 |
| 10 | | Dahla Dam Reservoir | 31.52 | 65.54 | 1070 | ---- | ---- | | ---- | 450 |


**Table 2:** Uniform prior and posterior distributions of model parameters for the calibrated models. The posterior column distributions show the parameter values of the best available parameter set as well as the 5/95[th] percentile of feasible solutions (in brackets). Note that loss factor $K_L$ had negligible influence and was thus set to 0 for the models of ID6 and ID7 to keep to the principle of model parsimony.

| ID | Sub-basin symbols | Parameter | Prior distribution | Posterior distribution |
|----|------|------|------|------|
| 1 | UHRB$_U$ | $I_{max}$ [mm] | 0-2 | 0.13 (0.11-0.55) |
| | | $C_e$ [-] | 0.2-1 | 0.44 (0.36-0.57) |
| | | $S_{umax}$ [mm] | 40-800 | 250 (112-550) |
| | | $\beta$ [-] | 0.2-3 | 1.08 (0.68-1.55) |
| | | $P_{max}$ [mm d$^{-1}$] | 0.009-1 | 0.67 (0.65-0.70) |
| | | $T_{lag}$ [d] | 2-7 | 3.12 (3.00-3.84) |
| | | $K_f$ [d$^{-1}$] | 0.01-0.1 | 0.07 (0.06-0.08) |
| | | $K_s$ [d$^{-1}$] | 0.0009-0.01 | 0.001 (0.001-0.002) |
| | | $T_{th}$ [°C] | -2.5 – 2.5 | -1.12 (-1.42 -0.69) |
| | | $F_{dd}$ [mm °C$^{-1}$ d$^{-1}$] | 0-3 | 0.38 (0.27-0.51) |
| 4 | UARB$_U$ | $I_{max}$ [mm] | 0-2 | 0.45 (0.10-0.83) |
| | | $C_e$ [-] | 0.2-1 | 0.84 (0.41-0.84) |
| | | $S_{umax}$ [mm] | 40-800 | 200 (100-430) |
| | | $\beta$ [-] | 0.2-3 | 1.73 (0.93-2.27) |
| | | $P_{max}$ [mm d$^{-1}$] | 0.009-1 | 0.47 (0.15-0.47) |
| | | $T_{lag}$ [d] | 2-7 | 2.41 (2.00-3.01) |
| | | $K_f$ [d$^{-1}$] | 0.01-0.1 | 0.07 (0.03-0.08) |
| | | $K_s$ [d$^{-1}$] | 0.0009-0.01 | 0.001 (0.001-0.003) |
| | | $T_{th}$ [°C] | -2.5 – 2.5 | -1.35 (-1.50 -1.14) |
| | | $F_{dd}$ [mm °C$^{-1}$ d$^{-1}$] | 0-3 | 0.85 (0.39-1.99) |
| 6 | LARB | $I_{max}$ [mm] | 0.1-3 | 1.66 (0.97-2.15) |
| | | $C_e$ [-] | 0.1-1 | 0.23 (0.18-0.33) |
| | | $S_{umax}$ [mm] | 40-600 | 455 (200-515) |
| | | $\beta$ [-] | 0.1-3.00 | 2.76 (1.56-2.82) |
| | | $P_{max}$ [mm d$^{-1}$] | 0.01-0.1 | 0.04 (0.03-0.05) |
| | | $T_{lag}$ [d] | 2-7 | 3.45 (2.12-4.18) |
| | | $K_f$ [d$^{-1}$] | 0.01-1.00 | 0.02 (0.01-0.02) |
| | | $K_s$ [d$^{-1}$] | 0.0009-0.01 | 0.009 (0.008-0.01) |
| | | $K_L$ [-] | 0.00 | 0.00 (0.00-0.00) |
| 7 | LHRB | $I_{max}$ [mm] | 0.1-3 | 1.58 (0.27-1.85) |
| | | $C_e$ [-] | 0.1-1 | 0.19 (0.11-0.35) |
| | | $S_{umax}$ [mm] | 40-600 | 515 (220-585) |
| | | $\beta$ [-] | 0.1-3.00 | 2.81 (1.86-2.88) |
| | | $P_{max}$ (mm/day) | 0.01-0.1 | 0.03 (0.02-0.05) |
| | | $T_{lag}$ [d] | 3-10 | 6.61 (3.42-7.12) |
| | | $K_f$ [d$^{-1}$] | 0.01-1.00 | 0.03 (0.02-0.05) |
| | | $K_s$ [d$^{-1}$] | 0.0009-0.01 | 0.009 (0.005-0.01) |
| | | $K_L$ [-] | 0.00 | 0.00 (0.00-0.00) |
| 8 | SISP | $I_{max}$ [mm] | 0.1-3 | 1.58 (0.27-1.85) |
| | | $C_e$ [-] | 0.1-1 | 0.19 (0.11-0.35) |
| | | $S_{umax}$ [mm] | 40-600 | 515 (220-585) |
| | | $\beta$ [-] | 0.1-3.00 | 2.81 (1.86-2.88) |
| | | $P_{max}$ [mm d$^{-1}$] | 0.01-0.1 | 0.03 (0.02-0.05) |
| | | $T_{lag}$ [d] | 3-10 | 6.61 (3.42-7.12) |
| | | $K_f$ [d$^{-1}$] | 0.01-1.00 | 0.03 (0.02-0.05) |
| | | $K_s$ [d$^{-1}$] | 0.0009-0.01 | 0.009 (0.005-0.01) |

| | | $K_L$ [-] | 0-1 | 0.34 (0.33 – 0.36) |
|---|---|---|---|---|
| | | $a_h$ [d$^{-1}$] | - | 0.27 (0.13-0.40) |
| | | $b_h$ [-] | - | 0.64 (0.50-0.77) |
| 9 | UHRB$_D$ | $c_h$ [mm d$^{-1}$] | - | -173 (-332- -15) |
| | | $a_l$ [d$^{-1}$] | - | 0.13 (0.09-0.17) |
| | | $c_l$ [mm d$^{-1}$] | - | 217 (173-262) |
| | | $a_h$ [d$^{-1}$] | - | 0.21 (0.11-0.31) |
| | | $b_h$ [-] | - | 0.86 (0.79-0.93) |
| 10 | UARB$_D$ | $c_h$ [mm d$^{-1}$] | - | -58 (-84- -32) |
| | | $a_l$ [d$^{-1}$] | - | 0.26 (0.11-0.42) |
| | | $c_l$ [mm d$^{-1}$] | - | 25 (17-33) |






**Table 3.** Classification of standardized drought indices DI used in this study (SPI, SPEI and SDI).

| Classification | DI [-] | Probability [-] |
|---|---|---|
| No drought | $DI > 0$ | 0.501 |
| Mild drought | $-1 \leq DI < 0$ | 0.341 |
| Moderate drought | $-1.5 \leq DI < -1$ | 0.092 |
| Severe drought | $-2 \leq DI < -1.5$ | 0.044 |
| Extreme drought | $DI < -2$ | 0.023 |


**Table 4.** Model performance metrics for calibration and validation in all study sub-basins. The values include the best performing model as well as the range of all solutions retained as feasible (in brackets)

| Location ID | Sub-basin Symbol | Calibration period (1971 – 1975) | | Validation period (1976–1979) | |
|---|---|---|---|---|---|
| | | $E_{NS,Q}$ | $E_{NS,log(Q)}$ | $E_{NS,Q}$ | $E_{NS,log(Q)}$ |
| 1 | UHRB$_U$ | 0.82 (0.82-0.83) | 0.91 (0.90-0.91) | 0.80 (0.79-0.80) | 0.86 (0.86-0.87) |
| 2 | UHRB$_D$ | 0.79 (0.78-0.80) | 0.81 (0.79-0.82) | 0.79 (0.79-0.80) | 0.85(0.84-0.86) |
| 4 | UARB$_U$ | 0.83 (0.83-0.84) | 0.85 (0.85-0.86) | 0.73 (0.72-0.73) | 0.78 (0.78-0.89) |
| 5 | UARB$_D$ | 0.89 (0.88-0.90) | 0.92 (0.91-0.92) | 0.74 (0.74-0.75) | 0.80 (0.79-0.81) |
| 6 | LARB | 0.70 (0.69-0.71) | 0.73 (0.71-0.74) | 0.81 (0.80-0.83) | 0.83 (0.81-0.86) |
| 7 | LHRB | 0.82 (0.81-0.83) | 0.85 (0.83-0.86) | 0.84 (0.82-0.86) | 0.88 (0.86-0.91) |
| | | (1971 – 1975) | | (1976 – 2006) | |
| 8 | SISP | 0.88 (0.86-0.89) | 0.89 (0.87-0.89) | 0.73 (0.68-0.74) | 0.75 (0.74-0.77) |


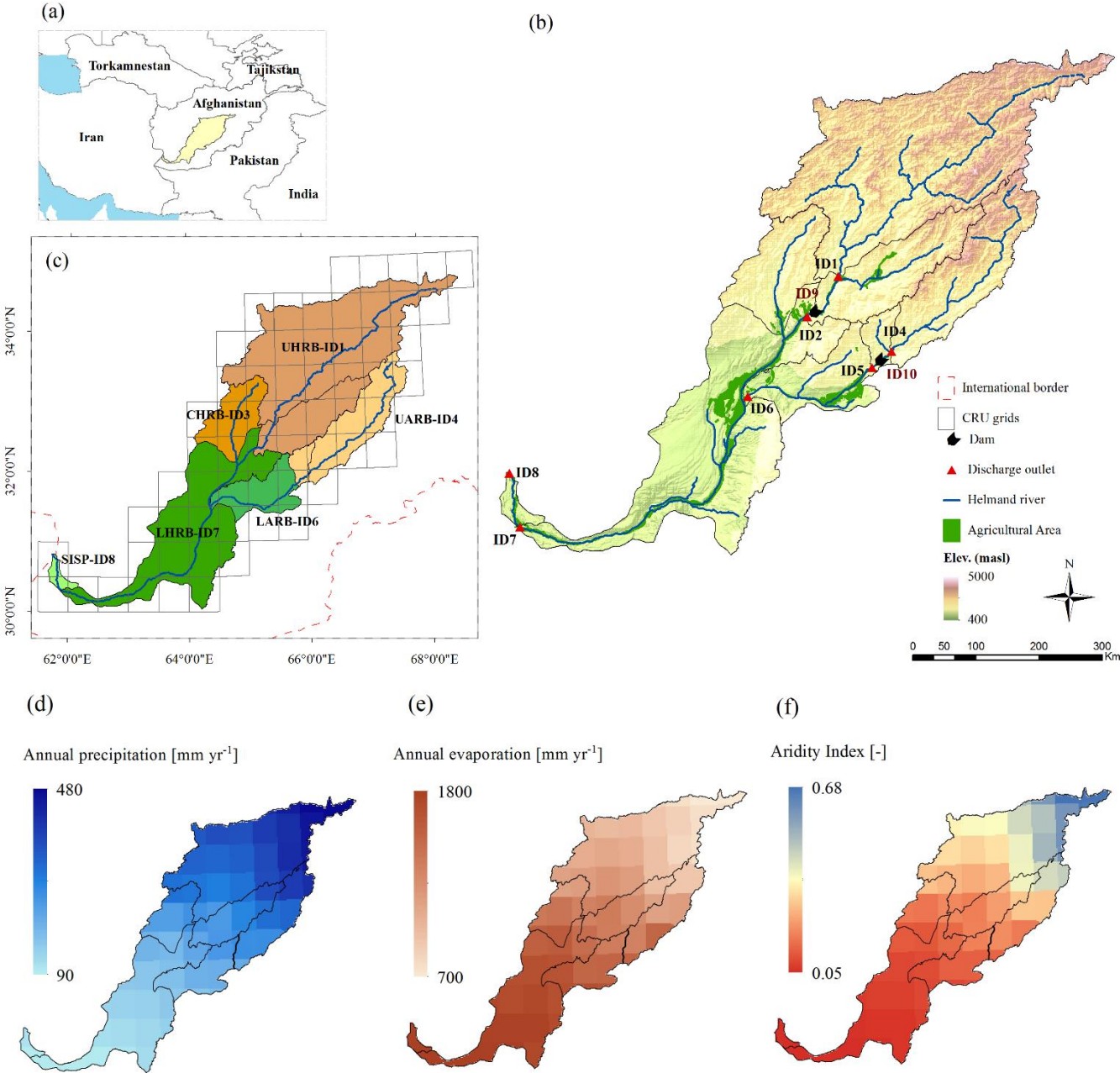

**Figure 1.** (a) The location of Helmand River Basin (HRB) in central Afghanistan, (b) elevation map of the HRB, also indicating the sub-basin boundaries, the locations of the sub-basin outlets and the agriculturally used area (as of 2006), (c) outline of the sub-basins analysed in this study, including the grid cells of CRU precipitation data used, (d) long-term mean annual precipitation P [mm y$^{-1}$], (e) long-term mean annual potential evaporation $E_P$ [mm y$^{-1}$] and (f) the aridity index $I_A$= P/$E_P$ [-]

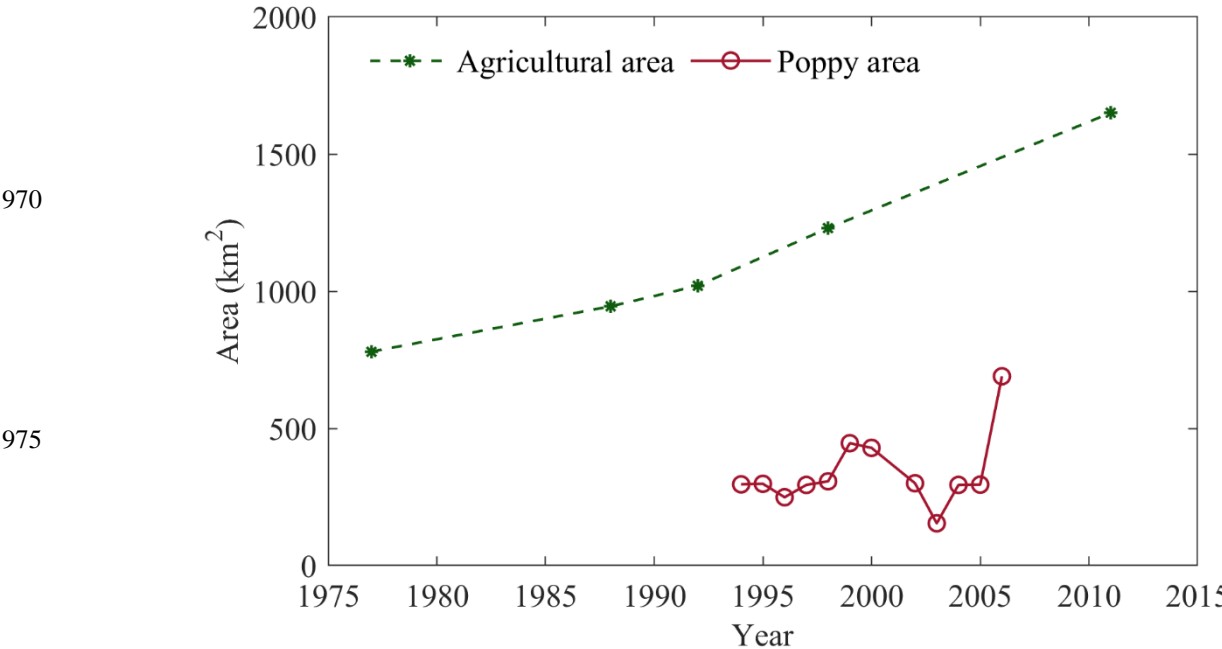

**Figure 2.** Evolution of total agricultural area in the HRB (1976-2011) and Poppy cultivated area thereof (1994-2006).

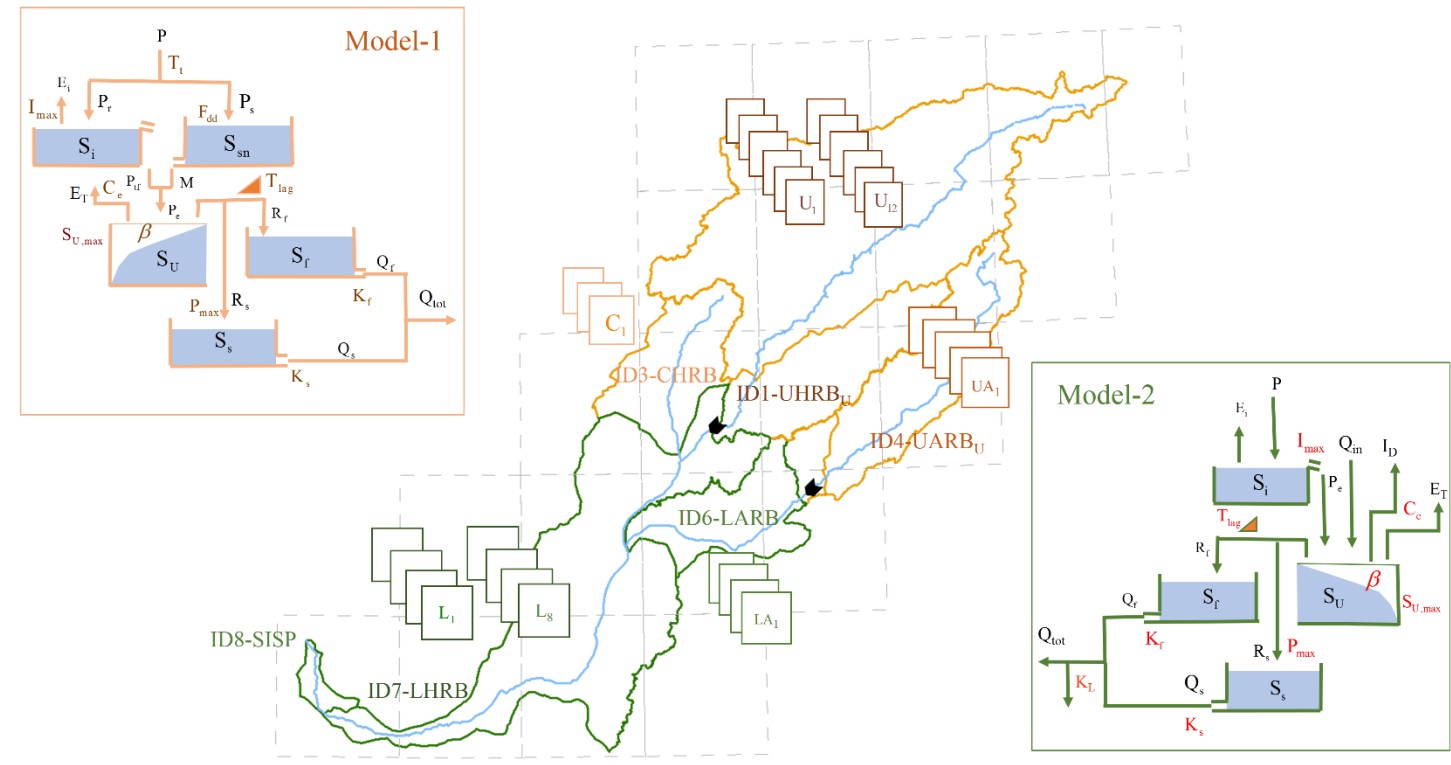

**Figure 3.** The distributed model structure consisting of parallel components (the structure of Model-1 used for UHRB, CHRB and UARB, Model-2 is used in LHRB, LARB and SISP) and 32 units of CRU grid cell, representing one subbasin each, characterized by an individual parameter set. Variables: P total precipitation [mm d$^{-1}$], P$_s$ snowfall [mm d$^{-1}$], P$_r$ rainfall [mm d$^{-1}$], M snowmelt [mm d$^{-1}$], P$_{tf}$ throughfall [mm d$^{-1}$], P$_e$ effective precipitation [mm d$^{-1}$], E$_T$ transpiration [mm d$^{-1}$], E$_i$ interception evaporation [mm d$^{-1}$], R$_f$ recharge of fast reservoir [mm d$^{-1}$], R$_s$ recharge of slow reservoir [mm d$^{-1}$], I$_D$ irrigation demand for LHRB and LARB, respectively., $Q_{in} = Q_{UHRB_U} + Q_{LARB} + Q_{CHRB}$ [mm d$^{-1}$], Q$_f$ runoff from fast reservoir [mm d$^{-1}$], Q$_s$ runoff from slow reservoir [mm d$^{-1}$], Q$_{tot}$ = total runoff [mm d$^{-1}$], S$_{sn}$ storage in snow reservoir [mm], S$_i$ storage in interception reservoir [mm], S$_U$ storage in unsaturated reservoir [mm], S$_f$ storage in fast reservoir [mm], S$_s$ storage in slow reservoir [mm]. Parameters: T$_t$ threshold temperature [℃], F$_{dd}$ melt factor [mm ℃$^{-1}$ d$^{-1}$], I$_{max}$ interception capacity [mm], S$_{U,max}$ storage capacity in unsaturated reservoir [mm], β shape parameter [-],P$_{max}$ percolation capacity [mm d$^{-1}$], Ce runoff generation coefficient [-], K$_f$ storage coefficient of fast reservoir [d$^{-1}$], K$_s$ storage coefficient of slow reservoir [d$^{-1}$], K$_L$ loss factor [-] and, T$_{lag}$ lag time [d].

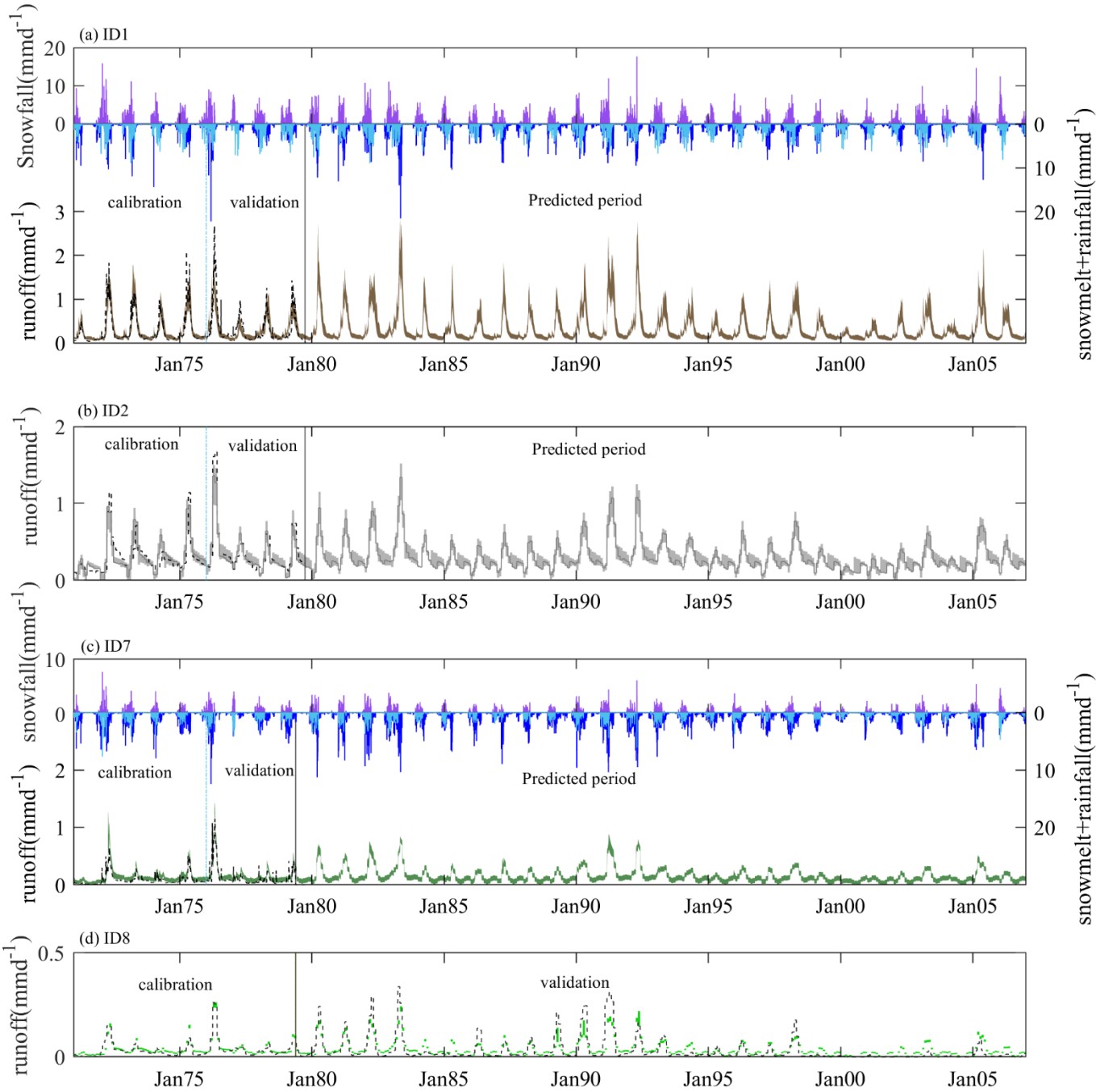

**Figure 4.** Precipitation and streamflow in UHRB$_U$ (ID1), UHRB$_D$ (ID2), LHRB (ID7), and SISP (ID8). The purple bars show the modelled snowfall P$_S$ [mm d$^{-1}$], the dark blue bars the modelled snowmelt M [mm d$^{-1}$] and the light blue bars the modelled rainfall P$_R$ [mm d$^{-1}$]. The dashed black lines indicate the observed runoff and the shaded areas the uncertainty ranges of modelled runoff during calibration, validation and prediction periods

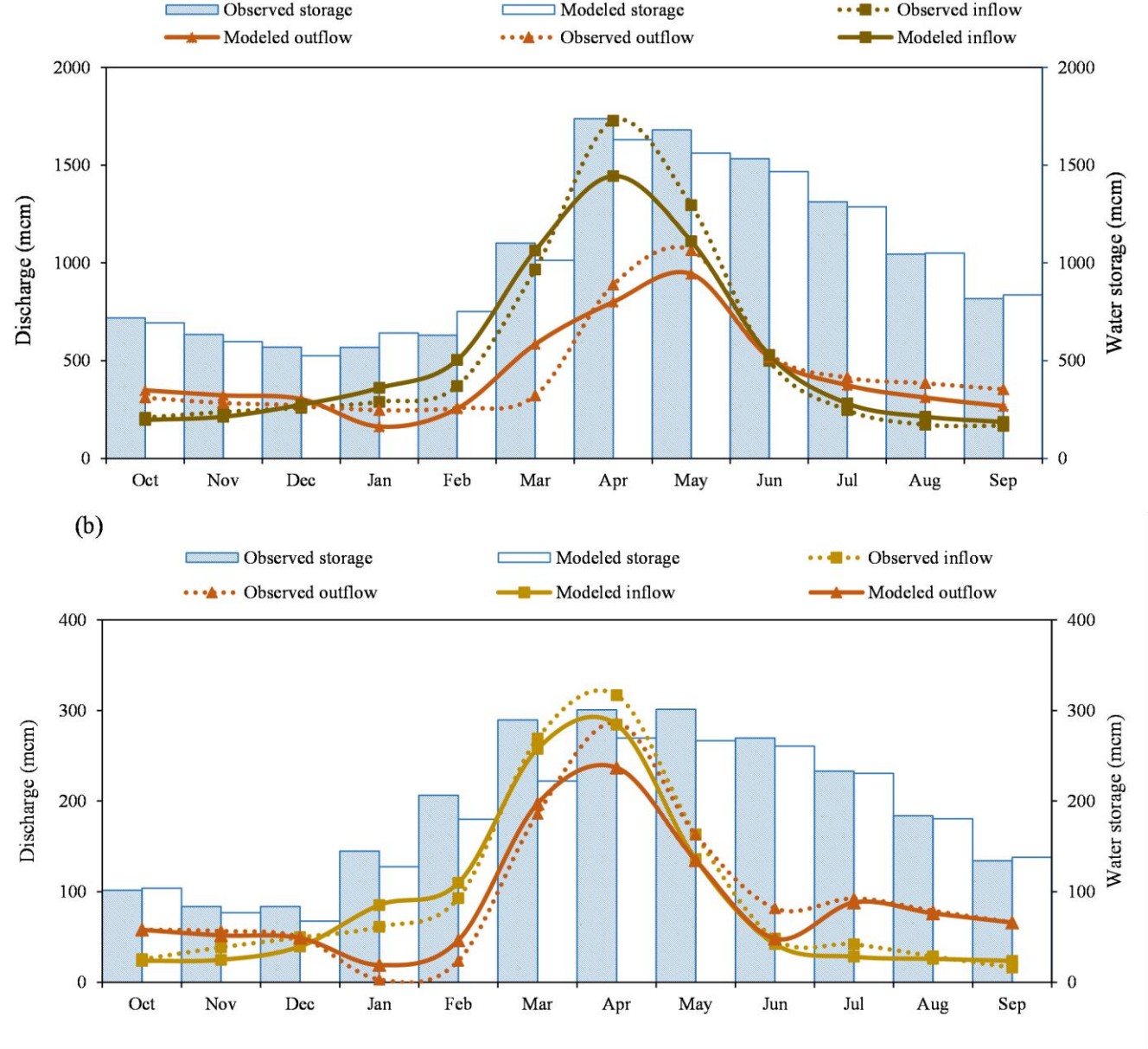

**Figure 5.** Mean observed and modeled inflow, outflow, and storage volume at (a) the Kajakai and (b) Dahla Dam reservoirs during the 1970–1979 period

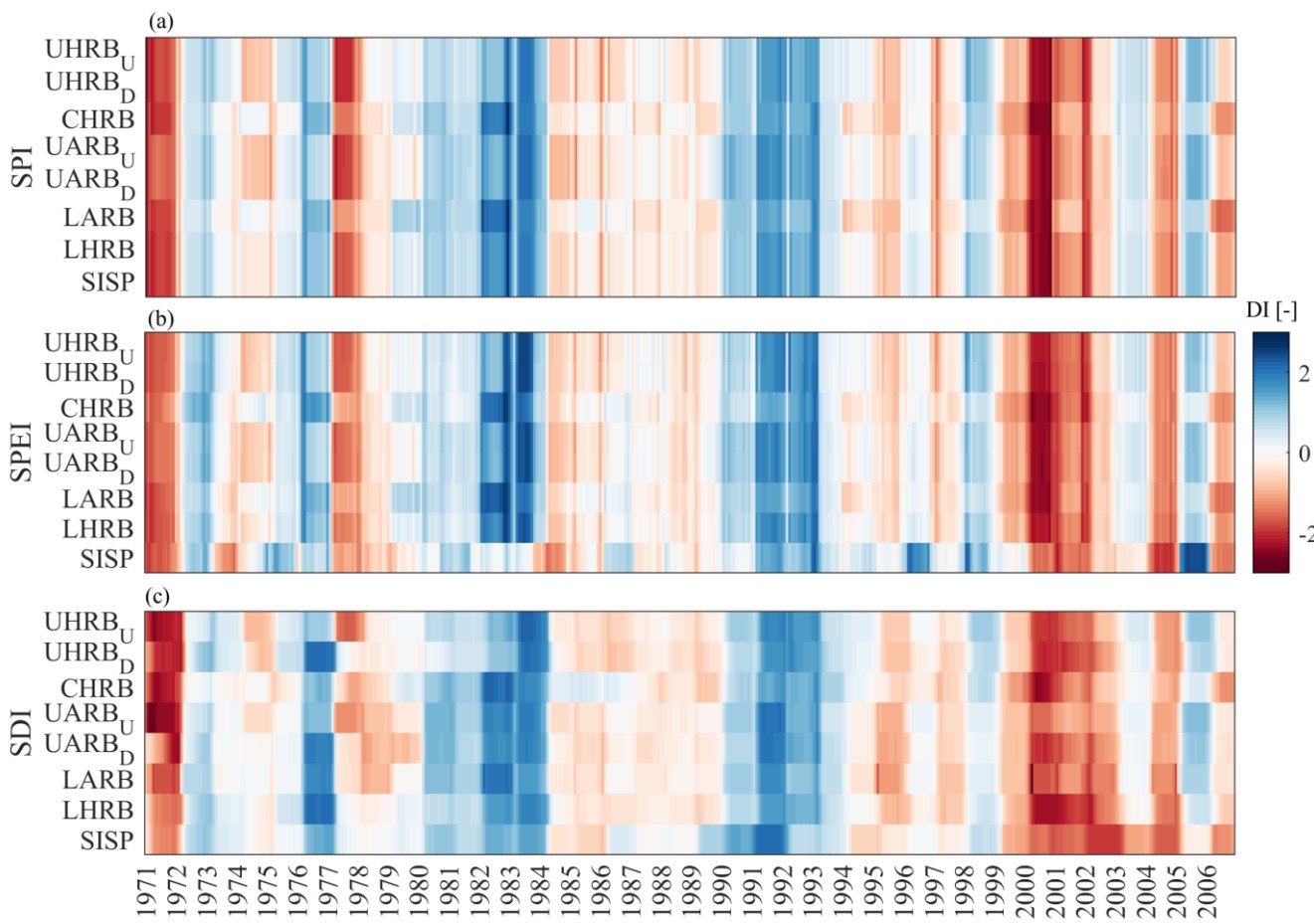

**Figure 6.** Time series of monthly drought indices (based on 12 months accumulation time) SPI, SPEI and SDI for the sub-basins ID1 – ID8 for the 1970 – 2006 study period

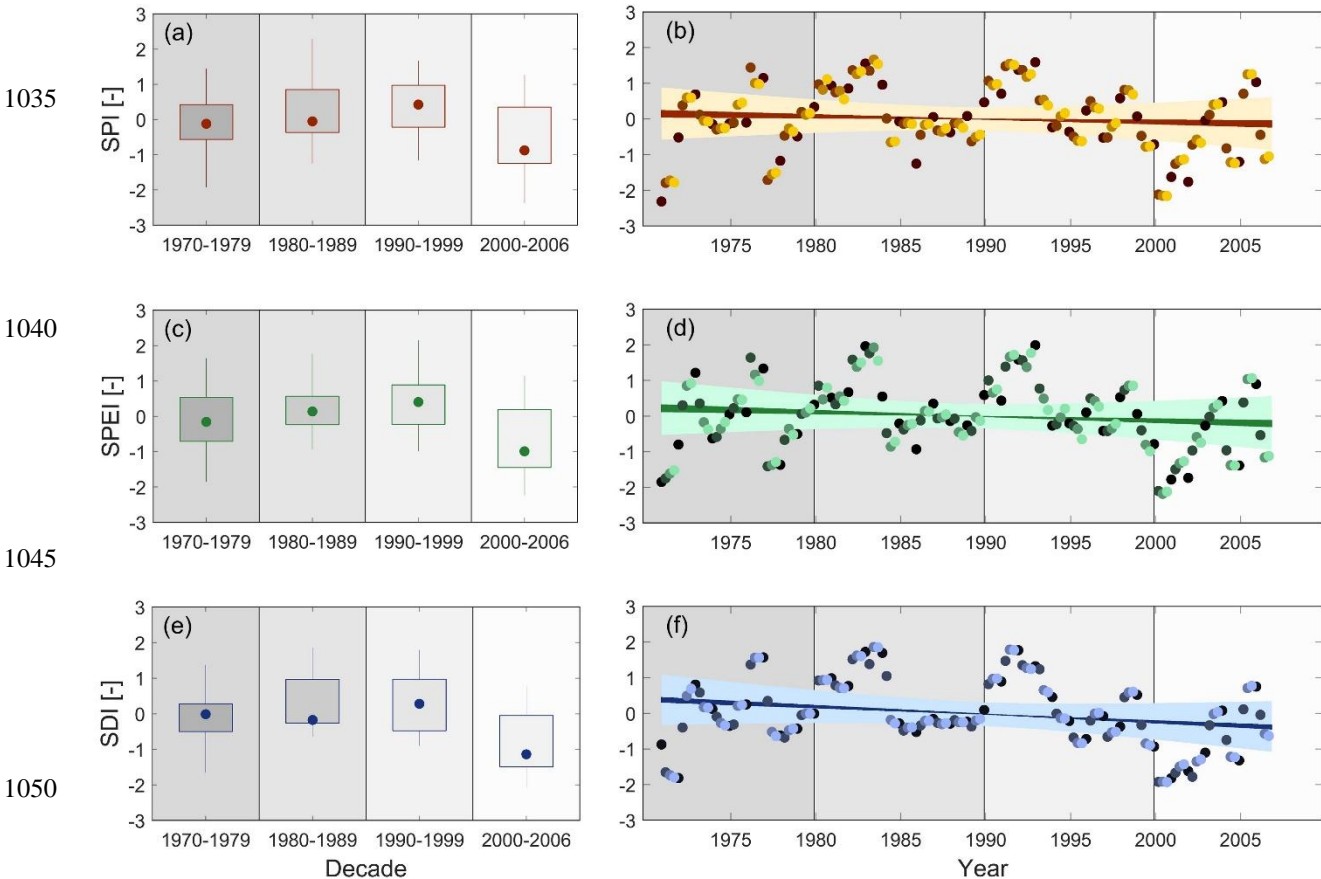

**Figure 7.** Decadal distributions and time series of mean basin (a)-(b) SPI, (c)-(d) SPEI and (e)-(f) SDI over the study period. The dots in the box plots indicate the median values and the whiskers the 5/95[th] percentiles. The dark to light shaded dots in the time series plots indicate the monthly drought indices (based on 12 months accumulation time) for all months of January, April, July and October, respectively. The dark shaded areas indicate the envelope of trend lines for the trends estimated based on all months of January, April, July and October, respectively. The light shaded areas show the associated envelope of 5/95[th] confidence intervals.

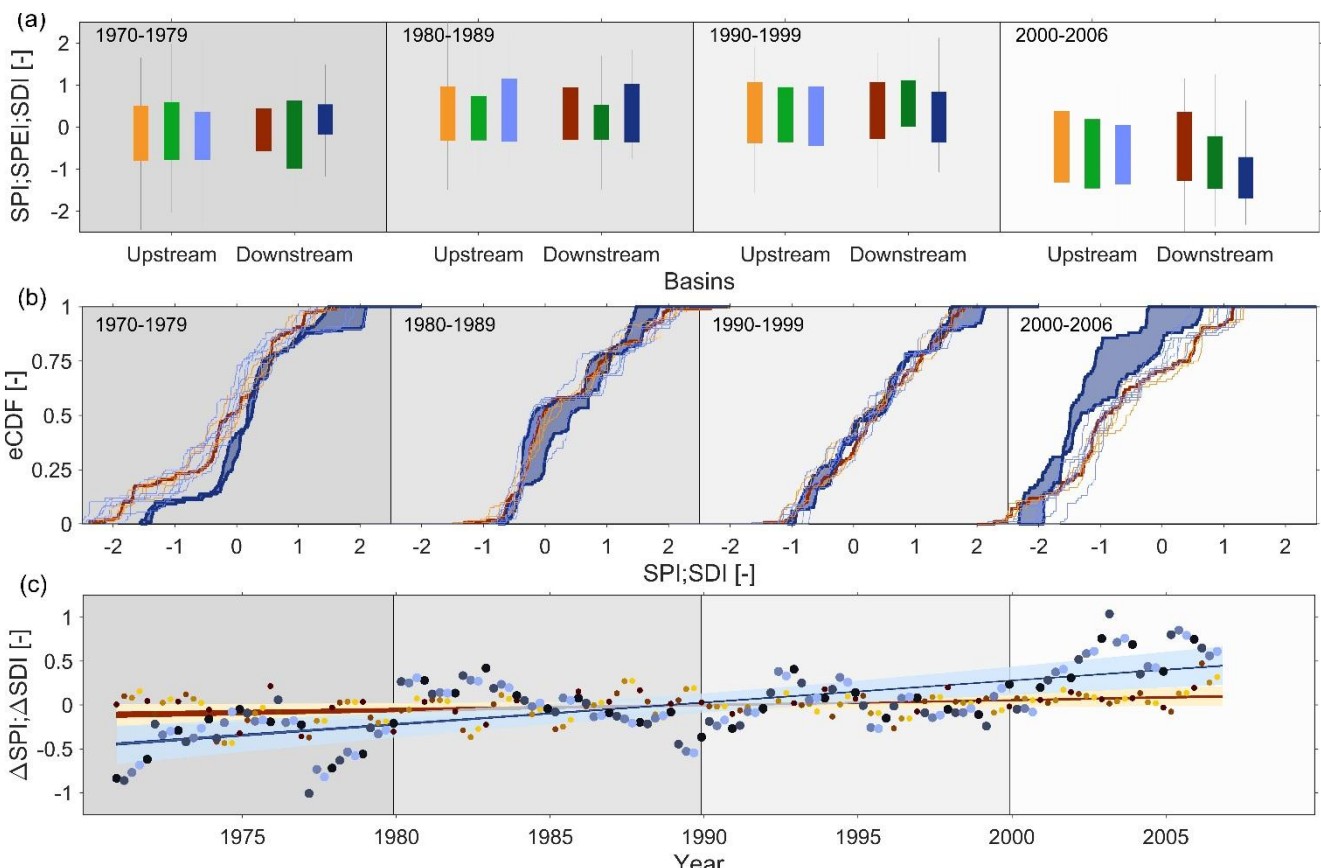

**Figure 8.** (a) decadal distributions of SPI, SPEI and SDI for the most upstream sub-basins (ID1-ID6) and the downstream sub-basins (ID7-ID8), (b) decadal empirical cumulative distribution functions of SPI (thins red lines upstream basins, bold red lines: downstream basins) and SDI (thin blue lines: upstream basins, bold blue lines: downstream basins). Note that the blue shaded area is added for better visualization of the shifts in downstream basins only and does not have a specific meaning. (c) time series of differences between mean upstream and mean downstream SPI (ΔSPI: yellow and red shades) as well as between mean upstream and mean downstream SDI (ΔSDI: blue shades). The symbols with shades from dark to light indicate the monthly SPI values (based on 12 months accumulation period) for the months January, April, July and October, respectively. The dark shaded areas indicate the envelope of trends in ΔSPI and ΔSDI, respectively, estimated based on all months of January, April, July and October, respectively. The light shaded areas show the associated envelope of 5/95[th] confidence intervals.

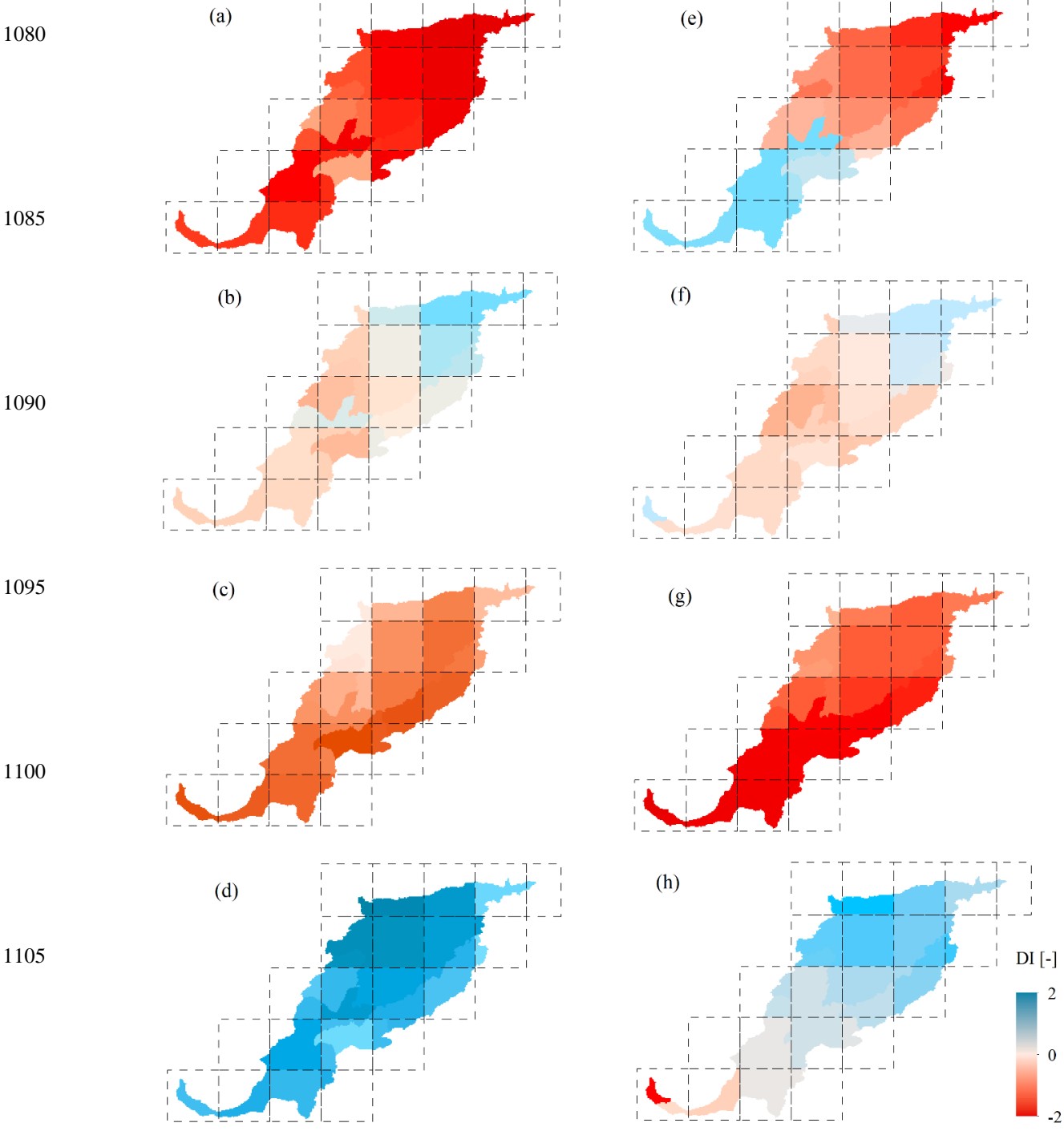

**Figure 9.** Spatial distribution of (a)-(d) SPI and (e)-(h) SDI for the years 1977, 1987, 2002 and 2003, based on the grid cells of the model application.

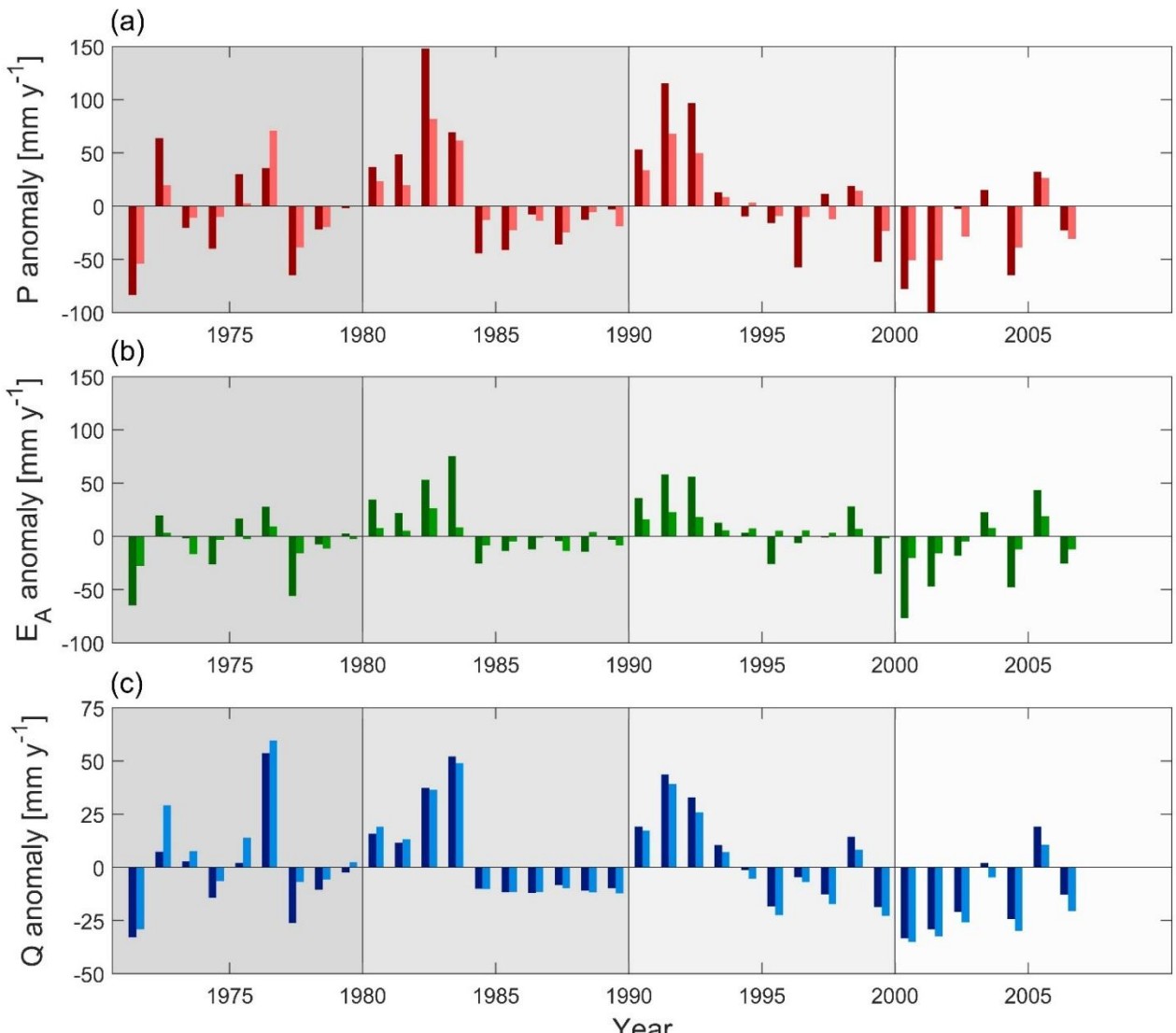

**Figure 10.** (a) Precipitation anomalies, (b) actual evaporation anomalies (here: $E_A = E_I + E_T$) and (c) streamflow anomalies over the study period. All anomalies are calculated based on the 1970-2006 mean values. The dark shaded bars indicate the combined flows to/from the upper basin (ID1-ID6), the light shaded bars show the flows to/from the lower basin (ID7).

1115

1120

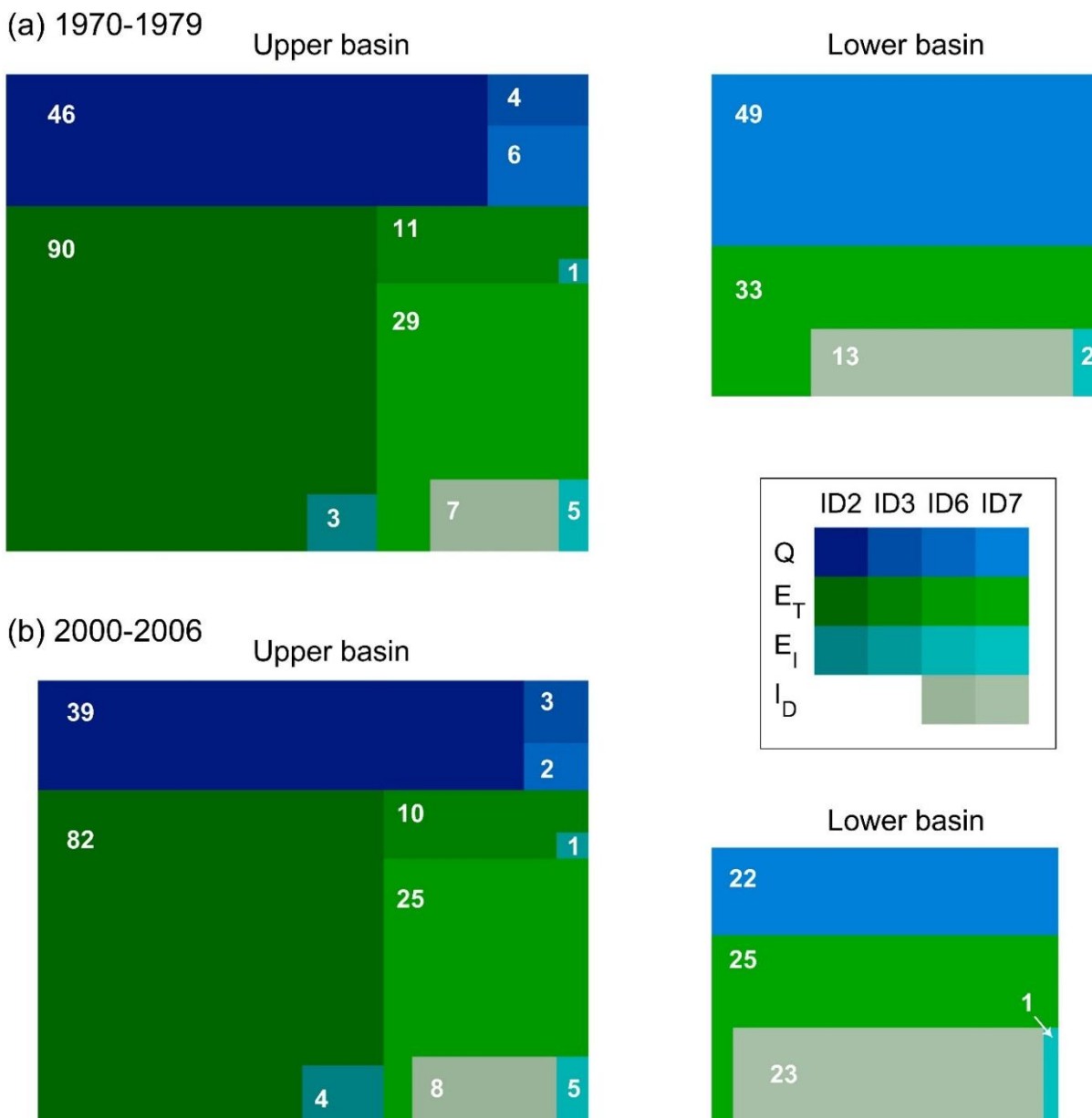

**Figure 11.** Water balances of the Upper and the Lower basins, respectively for (a) the 1970-79 and (b) the 2000-2006 periods. The size of the outer squares is equivalent to the total water available, i.e. for the upper basin precipitation P, for the lower basin precipitation P plus the combined inflow Q from the upper basin. The size of the internal rectangles of each flux (Q: streamflow, $E_T$: transpiration, $E_I$: interception evaporation and $I_D$: irrigation demand) in each sub-basin is equivalent to its fraction of the total available water in the upper and lower basin, respectively. The fluxes represent the decadal mean values and are shown in [mm y$^{-1}$].