# Peer review of "Signatures of human intervention – or not? Downstream intensification of hydrological drought along a large Central Asian River: the individual roles of climate variability and land use change"

_Hydrology and Earth System Sciences, 2020_

## Referee Comment (RC1) · Anonymous Referee #1 · 31 Aug 2020

The manuscript presents an interesting study on the intensification of downstream hydrological drought caused by meteorological and Human influences using a physically-based hydrological model and drought indices (SPEI, SPI and SDI). The study is suitable for publication in Hydrology Earth System Sciences (HEES) Journal. However, a major revision is required regarding the following comments;

General Comments 1. The introduction section needs to be rearranged and improved.

[Figure]

Specifically, the authors should start the introduction by stating how their research constitute a global problem before highlighting the regional problems identified in the Central Asian River Basin. I suggest that the authors start the introduction with line 66:81 (meaning it should be moved up and edited meaningfully) before discussion on the study basin. In doing so, however, the authors should ensure the literature cited are relevant in the body of work and should reflect in the discussion section. 2. The methods used in this research has a good scientific foundation. Nonetheless, the reasons for selecting the hydrological model need to be highlighted in a few sentences given that other hydrological models can perform this same task. Why use this model and why is it important. The authors mentioned the scarcity of data as a limitation of this research, does this model help to alleviate the problem of data scarcity? Why do the authors choose this model over other hydrological models? 3. Results and Discussion: The result was well presented; however, the discussion was not adequately presented. The authors need to discuss the result by comparing or contrasting conclusions made with relevant literature. The underlying physical processes and human activities that were outlined to be responsible for the derived result should have strong theoretical underpinning. Only then can the result have a strong scientific basis and meaning. If this is not done, this section may look like a mere presentation of result. I will suggest the authors separate the result from the discussion. It will help to know where the result presentation ends and where the discussion begins. 4. It is fascinating to see that the authors presented a limitation of this research. Clarity on limitations is beneficial in outlining the extent of possible errors. However, it will be more meaningful, for instance, if the authors specifically suggest a likely better approach to help limit uncertainties inherent in the result. For instance, is there a hydrological model that does better in the data-scarce region and that will capture different processes as highlighted by the model used; otherwise could developing such a model be a way to limit uncertainties? Moreso, will comparing different meteorological dataset to determine which best captures drought (with reference to SPEI and SPI) within the study site help limit uncertainty for better policy formulation on water resource management? All

these specific forward-looking documentation can be made to readers to know about possibilities and solutions since the data may inaccurately depict reality. And in-turn improve the quality of the manuscript 5. The grammar is sometimes not correct. The authors should allow a native speaker to help improve some sentences.

Specific comments Line 97: remove "s" from occurs Line 117-118: the sentence on this line ended with "most of the crops located in the traditionally irrigated areas (Wardlaw et al., 2013)". I feel this sentence is incomplete or rather add no additional meaning to the preceding sentence. Line 151: put "." After Ssn Line 161: delete "that" Line 261: remove "the" before give Line 294: Sentence not understandable. The authors placed Iran after requirement. The sentence needs to be rewritten to portray real meaning. Line 300-304: more explanation needs to be given about streamflow loses. Since soil evaporation is not enough reason as to why 60% of stream water is lost in the hyper-arid region, an explanation should be provided from the literature with regards to the inferences made by the author. Since the authors outlined that deep aquifer recharge and deep infiltration may be responsible for loses, more explanation should be given with references from existing literature. Line 350: delete "with" after was Line 355-359: there is a need to corroborate the role Ep plays in intensifying or moderating drought. The authors need to discuss with relevant literature. Line 434-436: The authors should compare or contrast their findings/conclusions with relevant literature Line 437: recast sentence. Line 441: remove "s" from estimate Line 486-487: the sentence starting with demand is not understandable, please recast. Line 500-501: what do the authors mean by "consequence of the extension the....." please recast to portray the intended meaning Line 508: Do not start the sentence with "this"

---

## Referee Comment (RC2) · Anonymous Referee #2 · 8 Oct 2020

Dear authors, I have read the manuscript "Signatures of human intervention – or not? (...)" with great interest, as I think the attribution of droughts to human or natural phenomena can be very interesting with regards to drought risk reduction policies. The study clearly states its hypotheses and is able to – with limited amount of info available-model droughts in the Helmand River Basin, including both hydrological and human components. The manuscript is well written, and I particularly do like the creative graphics and the nuance at the end of the paper. While I am satisfied with the general

setup and idea behind the analysis, I am left with a few concerns regarding the method, therefore I would like to recommend a major revision for this paper.

About the SETUP: The use of 10year periods to conclude about large trends is questionable. As droughts are (supposed to be) an extreme event, it is possible that some decades have more droughts than others without pointing to any climate- or human-related trend. Why not dividing it only in two periods? Or only looking at average trends?

About the RESERVOIR: A lot of your analysis of droughts is dependent on the assumption you make regarding the reservoir routing (line164+, in particular on line 175). The routing through reservoirs during low flow, in this case the most interesting one, has a rather low R2 (0.57). Have you done a sensitivity analysis to see how this affects your results? I think this should be more prominent in the discussion. Besides, there is the assumption of human reservoir operations that are absent, assuming the outflow is not adjusted by humans, but using the empirical link with the total storage distinguishing low and high flow, but make no distinction between drought and more-than-average-Q years. The conclusions about the influence of the reservoirs on the propagation of droughts should reflect this uncertainty.

About the MODEL: I feel I do not understand the additional parameters (such as deep infiltration losses) well: how are they parameterised? How do you know for sure this water is lost due to percolation? What is the importance and sensitivity of Snowmelt in the model? Since humans are not effective in applying irrigation water, there must be an underestimation of the water used for irrigation? I agree the end results of the hydrological bucket model are not bad (although the intra-annual variability is not very good), but with so many parameters, how sure are you that you model the correct processes? I suggest to add this to the discussion.

About the INDICES: I would like to see the goodness of fit of the gamma and GEV distributions for the SPI SPEI and SDI. They can matter a lot, a bad distribution (for

some months) could potentially affect the rest of your analysis, hence other distributions could be a solution. I was wondering if you used the Stagge et al 2015 approach to deal with zero values for the Gamma? Besides, I wonder why you would use an accumulation time of 12 months – in a very vulnerable environment as you work in, I would think an accumulation time of three months is more relevant, as the 12months can balance out dry and wet periods in the different seasons. I strongly suggest to try the same analysis with an accumulation time of 3months and see if your results still hold. Then you can indeed say you balance short and long term effects. Moreover, I do not understand why you would use a standardised value of 0 to determine a drought. Often -1 is used. Further, the whole analysis now only investigates below average conditions, that maybe not lead to any impacts. I would add the same analysis but for a threshold of -1 or -1.5, to see how real extremes change over time and through space. Again, this could really affect your conclusions. Finally, I also do not really understand how you include the lag time that usually exist between meteorological and hydrological droughts: it is logical that the SPI12Dec1987 is not consistent with the SDI12DEC1987 because droughts travel through the hydrological cycle with a certain lag time. Did you account for this?

About the DISCUSSION In the introduction (line50+), you cite a few authors who have started to analyse droughts in Afghanistan and the HRB, but you fail to explain what your approach will add to this. Also, did they find similar results ? It does not come back in your discussion sections (you cite a lot of numbers: how do they compare with other studies?).

Small comments related to the text: 1. In the abstract, I would specify human influence better. When you state "however the downstream parts of the HRB moderated the further propagation. . .." (l30) I would explain that this is because of the dams/reservoirs and/or land use – then it is easier to reflect on what caused the shift in this effect. Moreover, I would clearly state that you assume reservoirs without any human management. 2. In the introduction (line70+), you refer to Mishra and Singh, but this sentence is very

unclear. Please clarify that is the takeaway from this sentence. 3. I cannot find tis table S1 with al relevant model equations... (Only model variables) 4. Why would you show the actual instead of the potential evaporation in figure 10? SPEI uses potential, so that would reflect your drought analysis better.

---

## Author Comment (AC1) · 27 Nov 2020

We would like to thank the referee for the time and effort she/he put into this review. The points she/he raises are highly relevant and addressing them will help to improve our manuscript. We hope that with these replies and the associated revision of our manuscript all issues raised can be clarified.

[Figure]

Please also note the supplement to this comment:
https://hess.copernicus.org/preprints/hess-2020-369/hess-2020-369-AC1-
supplement.pdf

**Supplement:**

**Reply to Referee #1**

*Comment:*
*The manuscript presents an interesting study on the intensification of downstream hydrological drought caused by meteorological and Human influences using a physicallybased hydrological model and drought indices (SPEI, SPI and SDI). The study is suitable for publication in Hydrology Earth System Sciences (HEES) Journal.*

Response:
We would like to thank the referee for the time and effort she/he put into this review. The points she/he raises are highly relevant and addressing them will help to improve our manuscript. We hope that with these replies and the associated revision of our manuscript all issues raised can be clarified.

*Comment:*
*The introduction section needs to be rearranged and improved. Specifically, the authors should start the introduction by stating how their research constitute a global problem before highlighting the regional problems identified in the Central Asian River Basin. I suggest that the authors start the introduction with line 66:81 (meaning it should be moved up and edited meaningfully) before discussion on the study basin. In doing so, however, the authors should ensure the literature cited are relevant in the body of work and should reflect in the discussion section.*

Response:
We thank the Referee for having drawn our attention to this point and indeed we agree that it will help the logic and flow of the manuscript to start with a description of the general problem before going into regional detail. The text will be revised accordingly. Due to the scarcity of literature in this basin, we tried to cite the relevant studies in the study basin and other basins in similar environmental settings, subject to similar drought phenomena. In any case, we will extend the literature overview to provide some more context, in particular with respect to the global drought problematic.

*Comment:*
*The methods used in this research has a good scientific foundation. Nonetheless, the reasons for selecting the hydrological model need to be highlighted in a few sentences given that other hydrological models can perform this same task. Why use this model and why is it important. The authors mentioned the scarcity of data as a limitation of this research, does this model help to alleviate the problem of data scarcity? Why do the authors choose this model over other hydrological models?*

Response:
We agree that some more explanation on the choice of the model will help the reader to understand our decision. Briefly, the distributed implementation of this process-based model from the FLEX-model family was chosen as these models were previously successfully applied in climatically similar regions (e.g. Gao et al., 2014, 2017) but also many other settings worldwide (e.g. Fenicia et al., 2006; Kavetski et al., 2011; Nijzink et al., 2018; Bouaziz et al., 2018; Hulsman et al., 2020). In addition, the FLEX modelling concept is underlain by a philosophy of model customization and rigorous testing to ensure the implementation of suitable model formulations and the associated more reliable model outputs in different environments (e.g. Fenicia et al., 2011). Given the high number of different hydrological models that have been developed over the past decades, it is indeed plausible to assume that alternative model formulations could have been used to perform the same task. Please also note that, following a multi-objective calibration strategy, i.e.

simultaneously using $E_{NS,Q}$ and $E_{NS,log(Q)}$ as calibration objectives to ensure good representation of both, high- and low flows, our model performances with respect to daily flow in all sub-basins (Table 4, Figure 4), exceed those of the studies of Hajihosseini et al. (2016) but also those of Hajihosseini et al. (2019) who assessed the monthly flow with the SWAT model in the Upper and Lower Helamand basins, respectively. We will clarify this in the revised manuscript.

Scarcity of data is a problem that is faced by *any* type of model application in science and engineering. Available observation technology cannot provide us with the necessary observations to meaningfully quantify the natural heterogeneity of the (sub-)surface properties of large scale terrestrial hydrological systems, i.e. anything larger than the hillslope scale (e.g. Beven, 2001, 2006; Hrachowitz et al., 2014; Zehe et al., 2014). To obtain the parameters of hydrological models (which represent these properties) we therefore have to resort back to the calibration of inverse model applications. Given the number of parameters in these models, data scarcity frequently limits our ability to meaningfully constrain model parameters, thereby making the modelling problem ill-posed and resulting in the well-known equifinality problem (e.g. Beven and Binley, 1992; Savenjie, 2001; Beven, 2006; Clark et al., 2011; Hrachowitz and Clark, 2017). From that perspective, our model here does not "alleviate the problem of data scarcity". Rather, we have gone quite some length to extract as much information as possible from the available data to *constrain the parameters of our model* in this data-scarce environment to avoid the adverse effects of equifinality as much as possible.

However, what are our model results finally permitted us to do, within the margins of uncertainty associated to the equifinality problem, is to estimate stream flow in periods when no observations were available at some gauging stations in the study basin. From this alternative perspective, our model can indeed be considered to "alleviate the problem of data scarcity".

*Comment:*

*Results and Discussion: The result was well presented; however, the discussion was not adequately presented. The authors need to discuss the result by comparing or contrasting conclusions made with relevant literature. The underlying physical processes and human activities that were outlined to be responsible for the derived result should have strong theoretical underpinning. Only then can the result have a strong scientific basis and meaning. If this is not done, this section may look like a mere presentation of result. I will suggest the authors separate the result from the discussion. It will help to know where the result presentation ends and where the discussion begins.*

Response:

We acknowledge and appreciate this comment. However, given the stepwise nature of the analysis we would strongly prefer to have the results of the individual steps closely associated to the interpretation and discussion thereof. However, we will extend the discussion and provide some more detailed context and comparisons with the results of previous similar studies in the revised version of the manuscript.

*Comment:*

*It is fascinating to see that the authors presented a limitation of this research. Clarity on limitations is beneficial in outlining the extent of possible errors. However, it will be more meaningful, for instance, if the authors specifically suggest a likely better approach to help limit uncertainties inherent in the result. For instance, is there a hydrological model that does better in the data-scarce region and that will capture different processes as highlighted by the model used; otherwise could developing such a model be a way to limit uncertainties? Moreso, will comparing different meteorological dataset to determine which best*

*captures drought (with reference to SPEI and SPI) within the study site help limit uncertainty for better policy formulation on water resource management? All these specific forward-looking documentation can be made to readers to know about possibilities and solutions since the data may inaccurately depict reality. And in-turn improve the quality of the manuscript.*

Response:

We thank the Referee for the positive feedback on the limitation part of the study. As mentioned in one of the replies above, by customizing our model to the study basin and the data available there, we attempted to find an efficient trade-off between model complexity and model uncertainty. While we can of course not exclude that an alternative model formulation may do a better job, we still believe that we were able to strike a good balance between the process resolution and the resulting uncertainty in our model. This is in particular true as, after calibration for the 1971-1975 period, the daily model outputs for the gauge that recorded flow for the entire 1971-2006 period (ID8) remained rather good for the entire 35-year (!) study period (i.e. calibration period 1971-1975 and validation period 1976-2006; Table 4 and Figure 4). The best way to reduce uncertainties would be to do more observations and generate data, which could then be confronted with the model. Deficiencies in the model to reproduce this additional data could then, in an iterative process allow model improvement (e.g. Fenicia et al., 2008; Hulsman et al., 2020). However, at this stage, we believe that further model improvement will be difficult as the introduction of more complexity in the model will not be warranted by the actually available data and eventually merely lead to increased equifinality (Beven, 2006). We will clarify this in the revised manuscript.

We strongly agree with the reviewer that using alternative data sets, e.g. for precipitation, may lead to different results and interpretations. Although outside the scope of our manuscript, a future comparison of different data sets is therefore highly recommended. We will strongly emphasize this in the revised version of the manuscript.

*Comment:*

*The grammar is sometimes not correct. The authors should allow a native speaker to help improve some sentences.*

Response:

We will improve the grammar in the revised version.

*Comment:*

*Line 97: remove "s" from occurs.*

Response:

The word *occur* in this sentence refers to *precipitation*. Thus to the "s" is needed.

*Comment:*

*Line 117-118: the sentence on this line ended with "most of the crops located in the traditionally irrigated areas (Wardlaw et al., 2013)". I feel this sentence is incomplete or rather add no additional meaning to the preceding sentence.*

Response:

The sentence will be corrected to " … and large areas of opium poppy are grown especially in the traditionally irrigated area (Wardlaw et al., 2013)".

*Comment:*
*Line 151: put "."After Ssn.*
Response:
It will be corrected in the revised manuscript.

*Comment:*
*Line 161: delete "that".*
Response:
It will be corrected in the revised manuscript.

*Comment:*
*Line 261: remove "the" before give.*
Response:
It will be corrected in the revised manuscript.

*Comment:*
*Line 294: Sentence not understandable. The authors placed Iran after requirement. The sentence needs to be rewritten to portray real meaning.*
Response:
The sentence will be modified to "to meet irrigation demand in the downstream Helmand Valley and to satisfy the flow requirements of the Sistan River in Iran under the Iranian-Afghan Helmand River Water Treaty (1973)".

*Comment:*
*Line 300-304: more explanation needs to be given about streamflow loses. Since soil evaporation is not enough reason as to why 60% of stream water is lost in the hyperarid region, an explanation should be provided from the literature with regards to the inferences made by the author. Since the authors outlined that deep aquifer recharge and deep infiltration may be responsible for loses, more explanation should be given with references from existing literature.*
Response:
We would like to thank the Referee for raising our attention to this point. Addressing this comment we would like to clarify that we considered one additional parameter ($K_L$) to account for losses between ID7 (LHRB) and ID8 (SISP). It is true that we too confidently attributed these losses almost exclusively to deep groundwater export. Although this of course may play a role as in many other basins worldwide (e.g. Schaller and Fan, 2009; Bouaziz et al., 2018; Condon et al., 2020), we overlooked another even much more plausible source of these losses: when the Helmand River reaches Iran, it bifurcates just upstream the gauge at SISP (ID8) into the Sistan river (SISP, ID8), which drains into the Hamun wetlands, and the completely ungauged Common Parian River, which follows the border between Iran and Afghanistan. Therefore the lumped loss factor ($K_L$) combines the effects of deep infiltration, soil evaporation and particularly the proportion of water which is diverted into the Common Parian River. We will correct the definition of this parameter in the revised manuscript. Unfortunately there is not sufficient additional data available in the study area to directly estimate the losses. Merely Burger (2005) in a study of the Helmand River of Afghanistan and Iran loosely mentioned some schematization errors in their project including the existence of flow diversions from the Helmand River into the Common Parian River.

*Comment:*
*Line 350: delete "with" after was.*
Response:
It will be corrected in the revised manuscript.

*Comment:*
*Line 355-359: there is a need to corroborate the role Ep plays in intensifying or moderating drought. The authors need to discuss with relevant literature.*
Response:
Thank you for your suggestion. We will discuss the role of $E_P$ as an effective factor on droughts in the revised manuscript. However, please note that in arid environments such as in the study region, fluctuations in $E_P$ will have a limited effect on $E_A$ and thus on water deficits. As by definition, these regions are limited by water rather than by energy supply, $E_A$ will not significantly change when precipitation remains stable at e.g. 400 mm $yr^{-1}$ while $E_P$ fluctuates from e.g. 1000 mm $yr^{-1}$ in one year to 1300 mm $yr^{-1}$ in the following year. In either case, most of the 400 mm $yr^{-1}$ of available water will be evaporated as $E_A$. Changes in $E_P$ will therefore be less relevant for the intensification/moderation of drought in such arid regions than changes in precipitation.

*Comment:*
*Line 434-436: The authors should compare or contrast their findings/conclusions with relevant literature.*
Response:
We agree. In the revised manuscript we will extend the discussion and better place our results into a broader context of previous studies.
*Comment:*
*Line 437: recast sentence.*
Response:
We will adjust this sentence to "Such short-term influences of hydrological drivers deficits, would be likely to manifest themselves in the evolution of ΔSDI characterized by more erratic temporal pattern".

*Comment:*
*Line 441: remove "s" from estimate.*
Response:
It will be removed in the revised manuscript.

*Comment:*
*Line 486-487: the sentence starting with demand is not understandable, please recast.*
Response:
We will change the sentence in the revised manuscript to: "Irrigation demand in the lower basin accounts with $I_D \sim 13$ mm $y^{-1}$ for a substantially larger fraction of the water balance (~ 14%) than in the upper basin."

*Comment:*
*Line 500-501: what do the authors mean by "consequence of the extension the....." please recast to portray the intended meaning*

Response:
we will recast this sentence to " … is largely a consequence of increases in agriculturally used area which resulted in increases of the related irrigation water demand".

*Comment:*
*Line 508: Do not start the sentence with "this.*
Response:
It will be removed in the revised paper.

---

## Author Comment (AC2) · 27 Nov 2020

We appreciate Referee #2 for the very constructive feedback and detailed comments provided, which helped us to improve the manuscript. Please find the detailed responses to the individual comments below. We will incorporate these comments in the revised version of the manuscript. We hope that our responses to the reviewer comments clarify all issues raised.

Please also note the supplement to this comment:
https://hess.copernicus.org/preprints/hess-2020-369/hess-2020-369-AC2-
supplement.pdf

—————————————————
369, 2020.

**Supplement:**

**Reply to the comments from Referee #2**

*Comment:*
*Dear authors, I have read the manuscript "Signatures of human intervention – or not? (...)" with great interest, as I think the attribution of droughts to human or natural phenomena can be very interesting with regards to drought risk reduction policies. The study clearly states its hypotheses and is able to – with limited amount of info available model droughts in the Helmand River Basin, including both hydrological and human components. The manuscript is well written, and I particularly do like the creative graphics and the nuance at the end of the paper. While I am satisfied with the general setup and idea behind the analysis, I am left with a few concerns regarding the method, therefore I would like to recommend a major revision for this paper.*

Response:
We appreciate Referee #2 for the very constructive feedback and detailed comments provided, which helped us to improve the manuscript. Please find the detailed responses to the individual comments below. We will incorporate these comments in the revised version of the manuscript. We hope that our responses to the reviewer comments clarify all issues raised.

*Comment:*
*The use of 10year periods to conclude about large trends is questionable. As droughts are (supposed to be) an extreme event, it is possible that some decades have more droughts than others without pointing to any climate- or human-related trend. Why not dividing it only in two periods? Or only looking at average trends?*

Response:
We would like to thank the Referee for her/his comment. We strongly agree with the reviewer that droughts are extreme and thus by definition rare events that may extend over longer time scales. Individual decades may thus indeed exhibit varying occurrences of drought and to better understand climate- or human-induced effects on droughts longer time periods need to be considered. To do this and to limit the potential of misinterpretations, we therefore first analysed the long-term or "average" trends over the entire study period, as suggested by the reviewer and as shown in Figures 7b,d,f, 8c and 10 in the original manuscript. To provide the reader with more detail and to get a closer understanding of the underlying processes we then, in addition, zoomed in to the individual decades. This then allowed us to better analyse the overall trends and to see if there was indeed a systematic shift over time in the observed pattern or if the overall trends were merely artifacts of extreme events in the first or last decade of the study period. While the decadal considerations for the temporal analysis in section 5.2 and Figure 7 do not add substantially more information but merely support the overall trends, they provide very relevant and interesting insights in the analysis of the spatial drought pattern in section 5.3 of the original manuscript. This can, in particular, be seen in Figure 8b, where the decadal discretization reveals a clear gradual and systematic shift between upstream and downstream drought characteristics over the study period. We would therefore strongly prefer to keep the original decadal discretization as an addition to the long-term trends. In addition, we will add a Figure in the Supplementary Material showing the same analysis with a discretization of the study period into 2 periods only, as suggested by the reviewer, to illustrate also the longer-term validity of our interpretation.

*Comment:*
*A lot of your analysis of droughts is dependent on the assumption you make regarding the reservoir routing (line164+, in particular on line 175). The routing through reservoirs during low flow, in this case the most*

*interesting one, has a rather low R2 (0.57). Have you done a sensitivity analysis to see how this affects your results? I think this should be more prominent in the discussion. Besides, there is the assumption of human reservoir operations that are absent, assuming the outflow is not adjusted by humans, but using the empirical link with the total storage distinguishing low and high flow, but make no distinction between drought and more-than-average-Q years. The conclusions about the influence of the reservoirs on the propagation of droughts should reflect this uncertainty.*

Response:

We completely agree with the reviewer that a major source of uncertainty in the analysis is the lack of more detailed data on the operation of the two reservoirs in the Helmand River Basin. Although the storage-discharge relationship is statistically significant (p<0.001), the effect size remains modest. It is plausible to assume that reservoir operation is more careful during drier years. However, this should, at least to some degree, be implicit in the data we used to develop the reservoir routing scheme: during dry seasons water storage in the reservoirs is lower during drought periods than during wetter periods. Of course, this may not always apply. Although it may be a potentially very valuable idea to develop an individual reservoir operation scheme for drought periods only, we unfortunately do not have sufficient data to develop such relationships that are also sufficiently robust. We thus prefer to use all available data to inform a generally applicable scheme. Please also note that the estimated water release from the reservoirs results in overall model outputs in all downstream basins that are quite consistent with the observed daily river flow, which at station SISP (ID8) is even true for the entire 35-year study period. In spite of all other sources of uncertainty throughout the modelling process, this in itself can already be seen as an indication of the plausibility of the modelled reservoir outflow. However, we fully agree with the reviewer that a more detailed sensitivity analysis of the effects of the uncertainties in the reservoir routing scheme on downstream droughts will be very informative to better understand the limitations of or results. We therefore will add a detailed discussion on the effects of such a sensitivity analysis in the revised manuscript.

*Comment:*

*I feel I do not understand the additional parameters (such as deep infiltration losses) well: how are they parameterised? How do you know for sure this water is lost due to percolation? What is the importance and sensitivity of Snowmelt in the model? Since humans are not effective in applying irrigation water, there must be an underestimation of the water used for irrigation? I agree the end results of the hydrological bucket model are not bad (although the intra-annual variability is not very good), but with so many parameters, how sure are you that you model the correct processes? I suggest to add this to the discussion.*

Response:

We would like to thank the Referee for raising our attention to this point. Addressing this comment we would like to clarify that, in absence of further quantitative information, we parameterized these losses as a constant daily loss between stations LHRB (ID7) and SISP (ID8), as specified by the additional calibration parameter ($K_L$). We agree with the reviewer that we too confidently attributed these losses exclusively to deep groundwater export. Although this of course may play a role as in many other basins worldwide (e.g. Schaller and Fan, 2009; Bouaziz et al., 2018; Condon et al., 2020), we overlooked another even much more plausible source of these losses: when the Helmand River reaches Iran, it bifurcates just upstream the gauge at SISP (ID8) into the Sistan river (SISP, ID8) which then drains into the Hamun wetlands and into the completely ungauged Common Parian River, which follows the border between Iran and Afghanistan. Therefore the lumped loss factor ($K_L$) combines the effects of deep infiltration, soil evaporation and particularly the proportion of water which is diverted into the Common Parian River. We will correct the definition of this parameter in the revised manuscript. Unfortunately there is not sufficient additional data available in the study area to directly estimate the losses.

Snowmelt in the Helmand River Basin, mostly originating from the headwaters in the Upper Helmand (UHRB; ID1) and the Upper Arghandad River Basins (UARB; ID4), accounts for ~80% of river flow in the downstream sub-basins. As such, snowmelt is, by far, the most dominant source of river water in the region. The importance of the meltwater contribution then also leads to a well-constrained snowmelt model component as shown by rather narrow posterior distributions of the associated model parameters $T_T$ and $F_{dd}$ in Table 2 of the original manuscript.

We completely agree that the rather inefficient irrigation schemes in the study region may lead to an underestimation of the actual irrigation water use. With the available data, this is problematic to quantify in more detail, though. However, please note that modelled river flow reproduces the observed daily river flow at SISP (ID8) rather well over the entire 35-year study period, whereof the 30-year period 1976-2006 was a validation period without further model recalibration. Such a rather good model performance over a 30-year validation period is in itself an indicator that the hydrological dynamics of the system are reproduced by the model in a plausible way. In addition, if the modelled irrigation is an underestimation, it would nevertheless be a conservative estimate: higher real-world irrigation water demand would even further strengthen the conclusion of our analysis, that the shift form of downstream moderation to the intensification of hydrological drought over the study period is largely an effect of human intervention. We will clarify this with a more detailed discussion in the revised manuscript.

*Comment:*

*I would like to see the goodness of fit of the gamma and GEV distributions for the SPI SPEI and SDI. They can matter a lot, a bad distribution (for some months) could potentially affect the rest of your analysis, hence other distributions could be a solution. I was wondering if you used the Stagge et al 2015 approach to deal with zero values for the Gamma? Besides, I wonder why you would use an accumulation time of 12 months – in a very vulnerable environment as you work in, I would think an accumulation time of three months is more relevant, as the 12months can balance out dry and wet periods in the different seasons. I strongly suggest to try the same analysis with an accumulation time of 3months and see if your results still hold. Then you can indeed say you balance short and long term effects. Moreover, I do not understand why you would use a standardised value of 0 to determine a drought. Often -1 is used. Further, the whole analysis now only investigates below average conditions, that maybe not lead to any impacts. I would add the same analysis but for a threshold of -1 or -1.5, to see how real extremes change over time and through space. Again, this could really affect your conclusions. Finally, I also do not really understand how you include the lag time that usually exist between meteorological and hydrological droughts: it is logical that the SPI12Dec1987 is not consistent with the SDI12DEC1987 because droughts travel through the hydrological cycle with a certain lag time. Did you account for this?*

Response:

We would like to thank the Referee for her/his suggestion. We will add the goodness of fit of the gamma and GEV distributions for the drought indices in the revised manuscript.

Interesting point. As mentioned in section 4.2 of the original manuscript and also reiterated by the reviewer in the comment above, we used 12 months accumulation periods to estimate SPI and SDI. Thus we, fortunately, did not come into the situation that we had to deal with 0-values, as the 12-months precipitation and discharge were always non-zero. The choice of the accumulation period is indeed always delicate in drought studies. For that reason, we did a preliminary analysis on multiple accumulation periods, including 3, 6, 9, 12 and 24 months. From that analysis we believe that 12-months accumulation periods can strike a good balance between the shorter and longer-term effects. Of course, no accumulation period can fully describe all drought-related processes that occur across multiple temporal (and spatial) scales. Yet, we

chose the 12-months period here as we think that it can best deal with the time lag introduced by the snow accumulation and melt dynamics which are the dominant control on river flow in the study region. Overall the differences between the accumulation periods are minor and do largely not affect the relevant features of the observed pattern, as shown in the heatmaps of the 3 drought indices for the 3 and 12 months periods here below in Figures R1 and R2.

[Figure]

Figure R1. Time series of monthly drought indices (based on 3 months accumulation time) SPI, SPEI and SDI for the subbasins ID1 − ID8 for the 1970 − 2006 study period

[Figure]

Figure R2. Time series of monthly drought indices (based on 12 months accumulation time) SPI, SPEI and SDI for the subbasins ID1 − ID8 for the 1970 − 2006 study period

Similarly, the choice of a drought index threshold to define droughts remains rather subjective. We agree that many studies use a threshold of -1. We here chose a threshold of 0 to provide a more comprehensive picture, including all below-average conditions. However, please note that most and the most important part of our analysis (Figures 7 – 10) do not require the definition of any threshold. Instead, we directly analyse the distributions of the individual drought indices. By doing so we avoid the need to define a binary and somewhat arbitrary decision drought yes/no. Using the full distributions has several advantages: it does not only provide a more complete picture as it includes all observed conditions, but also eliminates any further subjectivity in the analysis. However, to bring our analysis closer to what is typically done, we will adjust our drought threshold to -1 and provide re-calculated estimates of drought frequency, duration, intensity and severity in section 5.2.

We also completely agree with the last point of the reviewer here. There can be time lags of several months between meteorological and hydrological droughts. We have previously analyzed this in a preliminary cross-correlation anaylsis, which suggested a typical time lag of 3-6 months between SPI and SDI droughts (see Figure R3 here below for LHRB, ID7). However, as in this manuscript we do not compare the propagation from SPI drought to SDI drought, but rather the downstream propagation of both types of drought individually, we decided not to show the analysis on the lag times between the two as it would divert the reader from the main analysis in our manuscript without actually adding value for the interpretation.

[Figure]

Figure R3. Heatmap of correlation coefficents from cross-correlation analysis between SPI and SDI for LHRB (ID7)

*Comment:*

*In the introduction (line50+), you cite a few authors who have started to analyse droughts in Afghanistan and the HRB, but you fail to explain what your approach will add to this. Also, did they find similar results? It does not come back in your discussion sections (you cite a lot of numbers: how do they compare with other studies?).*

Response:

Thank you for highlighting this important point. Addressing this comment we will provide a more detailed description of the novelty of our study and a more detailed discussion of our results in the context of previous studies which were conducted in the Helmand River Basin or similar environments. Briefly, most studies in the Helmand River Basin have so far remained limited to mere documentation and/or general

assessments of mostly meteorological drought characteristics. We here extended this scope also to SPEI and SDI droughts. We particularly focus on changes in drought characteristics over time and space and attempted to explicitly and quantitatively attribute these changes to climate and human activity.

*Comment:*
*In the abstract, I would specify human influence better. When you state "however the downstream parts of the HRB moderated the further propagation:" (l30) I would explain that this is because of the dams/reservoirs and/or land use – then it is easier to reflect on what caused the shift in this effect. Moreover, I would clearly state that you assume reservoirs without any human management.*
Response:
We thank the referee for these suggestions and we will adapt the text in the abstract accordingly in the revised version of the manuscript.

*Comment:*
*In the introduction (line70+), you refer to Mishra and Singh, but this sentence is very unclear. Please clarify that is the takeaway from this sentence.*
Response:
Thank you for raising our attention to this point. We will clarify the sentence and adjust it to: "As pointed out, amongst others, by Mishra and Singh (2010) the processes underlying droughts are complex because they are dependent on many interacting processes in terrestrial hydrological systems, such as the interaction between the atmosphere and the hydrological processes which feed moisture to the atmosphere.

*Comment:*
*I cannot find table S1 with al relevant model equations: (Only model variables)*
Response:
Table S1 (Water balance and constitutive equations used in FLEX model ) is in the supplementary materials of the paper.

*Comment:*
*Why would you show the actual instead of the potential evaporation in figure 10? SPEI uses potential, so that would reflect your drought analysis better.*
Response:
It is true that SPEI is typically based on potential evaporation $E_P$. However, we believe that in arid and thus water-limited regions fluctuations in $E_P$ are rather irrelevant compared to fluctuations in P or $E_A$. In other words, there will be little difference in the partitioning of water fluxes if under the same annual precipitation of e.g. 500 mm yr$^{-1}$, $E_P$ is 1000 or 1500 mm yr$^{-1}$, as in both cases actual evaporation $E_A$ will be close to (or even exceed) 400 mm yr$^{-1}$ and as therefore most of the available water will be evaporated. In contrast, more water will be evaporated as $E_A$ (even if $E_P$ remains stable) in years when more water is available and thus P is higher. By extension, the effects of evaporation on droughts in arid regions can only be meaningfully assessed by changes in $E_A$. That is also why we did not develop a strong link between SPEI and the $E_A$ anomaly in the text as analysed in section 5.4 and figure 10, where we tried to quantify the individual role of each component of the water balance for the development of hydrological drought in the HRB. We will clarify this in the revised manuscript.

**References:**

Schaller, M. F. and Fan, Y.: River basins as groundwater exporters and importers: implications for water cycle and climate modeling. J. Geophys. Res. Atmos. 114. http://dx.doi.org/10.1029/2008JD010636, 2009.

Condon, L. E., Atchley, A. L. and Maxwell, R. M.: Evapotranspiration depletes groundwater under warming over the contiguous United States. Nat. Commun. 11**,** 873. https://doi.org/10.1038/s41467-020-14688-0, 2020.

Bouaziz, L., Weerts, A., Schellekens, J., Sprokkereef, E., Stam, J., Savenije, H., and Hrachowitz, M.: Redressing the balance: quantifying net intercatchment groundwater flows, Hydrol. Earth Syst. Sci., 22, 6415–6434, https://doi.org/10.5194/hess-22-6415-2018, 2018.

Mishra, A. K. and Singh, V. p.: A review of drought concepts, J. Hydrol., 391, 202-216, https://doi.org/10.1016/j.jhydrol.2010.07.012, 2010.